# Towards Mitigating Factual Hallucination in LLMs through Self-Alignment with Memory

## Abstract

Despite the impressive performance of Large Language Models (LLMs) across numerous tasks and widespread application in real-world scenarios, LLMs still struggle to guarantee their responses to be accurate and aligned with objective facts. This leads to **factual hallucination** of LLMs, which can be difficult to detect and mislead users lacking relevant knowledge. Post-training techniques have been employed to mitigate this issue, yet they are usually followed by a trade-off between honesty and helpfulness, along with a lack of generalized improvements. In this paper, we propose to address it by augmenting LLM's fundamental capacity of leveraging its internal *memory*, that is, the knowledge derived from pre-training data. We introduce **FactualBench**, a comprehensive and precise factual QA dataset consisting of nearly 200k Chinese generative QA data spanning 21 domains for both evaluation and training purposes. Furthermore, we propose **self-alignment with memory**, i.e., fine-tuning the model via preference learning on self-generated pairwise data from FactualBench. Extensive experiments show that our method significantly enhances LLM's performance on FactualBench, with consistent improvements across various benchmarks concerning factuality, helpfulness and multiple skills. Additionally, different post-training techniques and tuning data sources are discussed to further understand their effectiveness.

## 1 Introduction

**Factual hallucination** occurs when large language models (LLMs) generate inaccurate or entirely fabricated information in response to user queries (Zhang et al., 2023b; Huang et al., 2023). Detecting and mitigating factual hallucinations is crucial, because such errors can undermine trust in these models and potentially cause significant harm to users, especially when they are used in high-stakes applications (Ji et al., 2023; Mello & Guha, 2023; Kornblith et al., 2022). However, identifying these hallucinations poses a unique challenge, as the fabricated content is often presented in a plausible and convincing manner, making it difficult for users and models to recognize inaccuracies (Kaddour et al., 2023; Zhang et al., 2023b). This complexity underscores the necessity of addressing factual hallucination as a critical focus for enhancing the reliability of LLMs (Tian et al., 2023).

Previous studies have explored mitigating hallucinations through post-training techniques such as Supervised Fine-tuning (SFT) (Elaraby et al., 2023; Moiseev et al., 2022; Santos et al., 2022) and Reinforcement Learning (RL) (Tian et al., 2023; Lin et al., 2024; Ouyang et al., 2022). However, the implementation of these methods can inadvertently introduce more hallucinations (Gudibande et al., 2023; Ji et al., 2023; Zhang et al., 2023b) if training on novel knowledge that model has not previously encountered (Gekhman et al., 2024; Huang et al., 2023), and create a trade-off between generating responses that are truthful and those considered helpful (Lin et al., 2024) when training under inappropriate signals, for example, balancing response length and factuality in long-form tasks or implicitly favoring specific response modes (Singhal et al., 2023; Sharma et al., 2024; Torabi et al., 2018; Kumar et al., 2024). These limitations restrict model's ability to generalize effectively to new tasks and domains (Zhang et al., 2023b; Huang et al., 2023).

These challenges motivate us to enhance model factuality and facilitate generalized improvements by more effectively leveraging existing *memory* (i.e. pre-trained knowledge), a fundamental capability of LLMs (Zhao et al., 2023), rather than injecting new information. Specifically, we concentrate on precise closed-book question-answering (QA) task, which serves as a metric for assessing a

model's ability to accurately utilize its stored knowledge (Roberts et al., 2020) and has a golden criterion: the correctness of the provided answer. However, current QA datasets are often insufficient and inaccurate for both comprehensive evaluation and enhancement of model performance.

To address the limitation, we propose **FactualBench**, a large-scale, multi-domain Chinese generative QA dataset designed to evaluate and improve knowledge utilization ability of LLMs. Factual-Bench is constructed from high-quality, publicly available encyclopedia entries that are commonly used in pre-training corpora (Liu et al., 2024b; Ando et al., 2024), ensuring its alignment with model's existing knowledge. It comprises 181,176 benign samples across 21 domains, representing a diverse set of important topics filtered by view counts. Preliminary evaluations with this dataset reveal that the task is challenging for most LLMs, despite not requiring advanced skills or long-tailed knowledge. Interestingly, we observe that models frequently generate correct answers under high-temperature configuration (Figure 1(a)), suggesting that the knowledge is internalized but not effectively utilized. This highlights a potential for improvement in factuality.

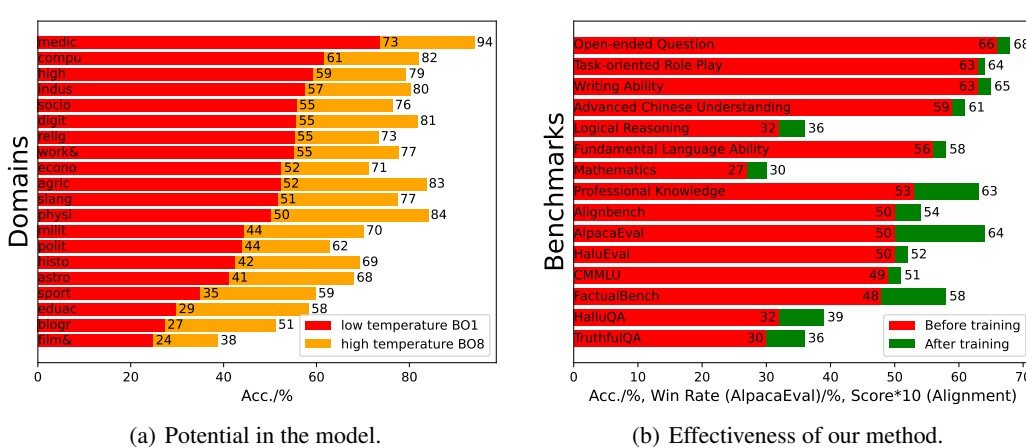

(a) Potential in the model.

(b) Effectiveness of our method.

Figure 1: Left: Model's performance on FactualBench. Sampling 8 answers in high temperature configuration, model is considered to have internalized the knowledge as long as one answer is correct. Orange bars indicate model's potential. Right: Model's perfomances on different benchmarks before and after training. The top eight tasks are sub-dimensions of AlignBench. Green bars indicate model's improvement after self-alignment with memory.

The potential suggests an inappropriate utilization of memory, which could be calibrated through alignment techniques. To unleash the potential of model for better factuality and generalized performance, we propose **self-alignment with memory**. Based on the train set of FactualBench that pertains to memory utilization, we elicit diverse responses from the model and use these self-generated outputs as labels rather than existing annotations to avoid training on their implicit answer modes. Subsequently we build pairwise data and post-train the model in preference learning (e.g., Direct Preference Optimization (Rafailov et al., 2024)) to deliver a more precise bi-directional signal.

We evaluate model using our test set and benchmarks that assess various dimensions and abilities, including CMMLU (Li et al., 2023a) for multiple-choice, HaluEval (Li et al., 2023b) for hallucination detection, TruthfulQA (Lin et al., 2022) and HalluQA (Cheng et al., 2023) for adversarial robustness, and AlpacaEval (Li et al., 2023c) for helpfulness, AlignBench (Liu et al., 2023) for comprehensive abilities. As illustrated in Figue 1(b), our method achieves a unanimous and significant increase in performance. We further discuss how our approach attains this effect from the relationship between memory and representation. We also conduct a series of experiments to investigate how different post-training algorithms and sources of tuning data influence the training outcomes.

## 2 RELATED WORKS

Hallucination is defined as generated content that is nonsensical or unfaithful to the provided source content (Ji et al., 2023; Huang et al., 2023). It presents a mutifaceted challenge Li et al. (2024b)

across a wide range of tasks (Ji et al., 2023). Hallucination can be categorized into intrinsic and extrinsic parts, depending on whether the response conflicts with context itself or with the fact beyond the context (Ji et al., 2023; Huang et al., 2023), where the latter type tends to have more side effects (Zhang et al., 2023b) and is mainly focused in our work.

To mitigate hallucinations, several studies (Gardent et al., 2017; Wang, 2019) find that enhancing the quality of pre-training data can be effective. But processing vast scale training data could be time-consuming and is not be applicable to models that complete pre-training. Other approaches focus on improving model's factuality through decoding strategies (Zhang et al., 2023a; Li et al., 2024b; Lee et al., 2022; Chuang et al., 2023), yet these strategies often increase inference complexity and have more difficulty generating fluent or diverse text (Ji et al., 2023). Additionally, some methods utilize retrieval-augmented generation techniques (Nakano et al., 2021; Gou et al., 2023), but introduce significant system complexity (Tian et al., 2023) and rely on external resources. Consequently, we emphasize enhancing model factuality through post-training by directly optimizing model's inherent parameters. Post-training algorithms, such as SFT and RL are frequently used to improve model's instruct following ability and align model with human preferences (Xu et al., 2024; Lin et al., 2024). However, SFT is considered to be sub-optimal for mitigating hallucinations due to its limited generalization capabilities in out-of-distribution cases (Zhang et al., 2023b). Recent RL studies improve factuality within a single task (Kang et al., 2024; Tian et al., 2023), or face a trade-off between factulity and helpfulness (Lin et al., 2024). In contrast, our method achieves generalized improvements across factuality, helpfulness, and comprehensive abilities.

The principle underlying our method is to enhance model factuality by better utilizing existing memory. However, current datasets are often insufficient and inaccurate for evaluating and improving this capability. Tasks that require long-form answers and multi-sentence reasoning (Wei et al., 2024; Min et al., 2023; Joshi et al., 2017; Yang et al., 2018) are often imprecise in measuring a model's fundamental knowledge utilization, as their performance is heavily influenced by instruct following and complex reasoning abilities. Similarly, multiple-choice and detection tasks that rely on rule-based automatic metrics (Li et al., 2023a;b; Liu et al., 2022; Mishra et al., 2024; Thorne et al., 2018) introduce significant biases into model evaluations (Lou et al., 2024). Datasets designed with adversarial intent, such as TruthfulQA (Lin et al., 2022) and others (Cheng et al., 2023), effectively stimulate factual hallucinations but tend to focus on specific scenarios, thereby limiting their capacity to reflect model accuracy on general, everyday questions. While datasets focused on precise generative QA (Yang et al., 2015; Wang et al., 2023a; Li et al., 2024a; Yin et al., 2023; Berant et al., 2013; Kwiatkowski et al., 2019) exist, they are generally constrained by small sample sizes or lack of domain classification, rendering them inadequate for a comprehensive evaluation of modern LLMs. These gaps in existing datasets motivate the need for more robust and large-scale resources like FactualBench, which we propose to address these shortcomings.

## 3 METHOD

As previously discussed, we aim to mitigate model's factual hallucination while achieving generalized improvements in helpfulness and comprehensive abilities. Prior research indicates that training on existing knowledge and accurate signals is essential for optimal training outcomes. Therefore we select precise QA task, judged solely by correctness, as training task to enhance model's fundamental capability of leveraging existing memory from pre-trained corpus. The optimization goal can be mathematically formalized as follows:

$$\max_{\pi_\theta} \mathbb{E}_{\mathbf{x} \sim \mathcal{X}^{\text{Fact}}, \, \mathbf{y} \sim \pi_\theta(\cdot|\mathbf{x})}[\mathcal{J}(\mathbf{x}, \mathbf{y})], \tag{1}$$

where $\mathbf{x}$ represents a factual question drawn from $\mathcal{X}^{\text{Fact}}$ space. This space encompasses straightforard questions about factual information, which can be grounded to reliable sources, including but not limited to dictionaries, encyclopedias, textbooks from different domains (Wang et al., 2023b). In our study, $\mathbf{x}$ is selected to exclude any malicious or misleading content, and the model's response $\mathbf{y}$ is generated solely from question and model's internal parameters without external references. We utilize a dataset $\mathbb{D}_{\text{Test}}$ to approximate $\mathcal{X}^{\text{Fact}}$. $\mathcal{J}$ functions as an evaluation metric that outputs either 0 or 1 based on the correctness of answer $\mathbf{y}$ to the question $\mathbf{x}$.

To achieve our goal, two key sub-questions should be answered: 1) How to generate a sufficient large and effective dataset to benchmark and enhance model performance on factual QA task? 2) What

methods can be employed to train the model to minimize factual hallucination, while simultaneously improving generalized capabilities?

## 3.1 FACTUALBENCH

A large-scale and comprehensive dataset containing benign and precise questions is needed for accurately evaluating and enhancing the knowledge utilization ability of models. We choose publicly available Internet encyclopedias as a reliable knowledge base as they are widely used as training corpora for LLMs (Liu et al., 2024b; Ando et al., 2024) and contain a wide range of topics across various domains (Bai et al., 2024). A model-based approach is adopted to generate large amounts of data costlessly and quickly. To avoid generating low-quality data (more details of such low quality data can be found in Appendix A), we utilize few-shot prompts (Brown, 2020), chain-of-thought (Wei et al., 2022) technologies, and apply several filtering strategies. GPT4 (Achiam et al., 2023) and Baichuan model[1] are adopted in construction for their strong command-following capabilities.

### 3.1.1 CONSTRUCTION AND COMPOSTION

The construction pipeline for FactualBench is organized into 5 steps: **1) Entry filtering**. We initially sampled millions of entries from encyclopedia data, covering a broad spectrum of subjects and domains. To avoid testing on long-tailed knowledge, we set a view-counts threshold, and 89,658 encyclopedia entries remained. **2) Description filtering**. Model's performance tends to decline as context length increases (Liu et al., 2024a; Sun et al., 2023; Li et al., 2024c). Excessively lengthy description may lead to low quality responses. Conversely, a overly brief description lacks sufficient factual information. To balance this, we filter out descriptions shorter than 100 characters and truncated those exceeding 800 characters. 64315 entries remained after this process. **3) Question generation**. We instruct GPT4 to generate at most 3 precise questions based on each truncated description. For each question, GPT4 is also required to provide 1 standard answer and 3 misleading incorrect answers. To ensure adherence to our instructions, we included two examples in the prompt as few shots. Totally 192,927 QA data are generated. **4) Question classification**. Following question generation, a domain classifier, fine-tuned on Baichuan model, is employed to categorize all questions into 20 distinct domains. Questions that don't fit into any domain are uniformly classified as *others*. **5) Question filtering**. We query GPT4 once more to filter out low-quality questions. Each question is assessed without corresponding encyclopedia description and GPT4 is instructed to judge whether question is low-quality through a step-by-step reasoning. Finally 181,176 questions are reserved, among which high-quality data accounts for 90% under human assessment. The complete prompts utilized throughout the generation pipeline can be found in Appendix B.1.1. Table 3 presents a sample from our dataset, while additional examples are provided in Appendix B.2.

Table 1: Samples distribution of Factualbench.

| Domain | 中文名 | Test | Training | Total |
|---|---|---|---|---|
| film&entertainment | 影视娱乐 | 201 | 54489 | 54690 |
| eduaction&training | 教育培养 | 161 | 3703 | 3864 |
| physics, chemistry, mathematics&biology | 数理化生 | 201 | 9189 | 9390 |
| history&traditional culture | 历史国学 | 202 | 18108 | 18310 |
| biography | 人物百科 | 201 | 11844 | 12045 |
| politics&law | 政治法律 | 175 | 6368 | 6453 |
| economics&management | 经济管理 | 160 | 4543 | 4703 |
| computer science | 计算机科学 | 201 | 6253 | 6454 |
| medical | 医学 | 167 | 7073 | 7240 |
| sociology&humanity | 社会人文 | 199 | 8503 | 8702 |
| agriculture, forestry, fisheries&allied industries | 农林牧渔 | 153 | 3728 | 3881 |
| astronomy&geography | 天文地理 | 160 | 3896 | 4056 |
| sports&tourism | 运动旅游 | 157 | 4869 | 5026 |
| digital&automotive | 数码汽车 | 176 | 3887 | 4063 |
| industrial engineering | 工业工程 | 172 | 3283 | 3455 |
| military&war | 军武战争 | 151 | 2569 | 2720 |
| slang&memes | 网词网梗 | 151 | 529 | 680 |
| work&life | 工作生活 | 174 | 5853 | 6027 |
| high technology | 高新科技 | 150 | 310 | 460 |
| religion&culture | 信仰文化 | 150 | 510 | 660 |
| others | 其他 | / | 18207 | 18207 |
| total | / | 3462 | 177714 | 181176 |

Table 2: Models performances on Factual-Bench rated by GPT4.

| Model | Proficient lang. | Acc. |
|---|---|---|
| Baichuan1 | CN | 48.24 |
| Baichuan2 | CN | 55.37 |
| Qwen1.5-7B | CN | 48.87 |
| Qwen2-7B | CN | 56.27 |
| Llama-3-8B | EN | 39.11 |
| Baichuan3 | CN | 67.50 |
| Yi-34B | CN | 67.30 |
| Command-R | EN | 54.30 |
| Llama-3-70B | EN | 49.65 |
| Qwen2-72B | CN | 73.71 |
| Baichuan4 | CN | 75.07 |
| Command-R+ 104B | EN | 60.17 |
| DeepSeek-v2-0627 MoE-236B | CN | 75.62 |
| GPT4-0125-preview | EN | 65.71 |

To establish an efficient benchmark, we randomly select 3,462 samples as the test set, while the remaining 177,714 samples comprised the training set. This selection is based on encyclopedia

---

[1] https://www.baichuan-ai.com/

entries instead of questions, ensuring all data in test set seperate from training set. Each domain has a similar number of questions in test set and entries containing *others* domain questions are excluded from selection. We manually rephrase unclear questions to maintain the quality of test set. The distribution of FactualBench is presented in Table 1.

Table 3: Each sample of FactualBench contains a question, a corresponding standard answer, three misleading incorrect answers and domain it belongs to. English translation is for reference only.

| Question | 第一台微波量子放大器是在哪一年制成的？ | In which year was the first microwave quantum amplifier made? |
|---|---|---|
| Standard Answer | 第一台微波量子放大器是在1954年制成的。 | The first microwave quantum amplifier was made in 1954. |
| Wrong Answer1 | 第一台微波量子放大器是在1958年制成的。 | The first microwave quantum amplifier was made in 1958. |
| Wrong Answer2 | 第一台微波量子放大器是在1960年制成的。 | The first microwave quantum amplifier was made in 1960. |
| Wrong Answer3 | 第一台微波量子放大器是在1962年制成的。 | The first microwave quantum amplifier was made in 1962. |
| Domain | 高新科技 | high technology |

### 3.1.2 EVALUATION

Similar to previous works (Liu et al., 2023; Zheng et al., 2024), a robust model-based approach is employed to expedite the assessment process. Given that rule-based automatic metrics such as ROUGE (Lin, 2004) and BLEU (Papineni et al., 2002) exhibit significant biases in evaluation (Lou et al., 2024), we assess the correctness of answer at semantic-level. The question, model answer, and standard answer are provided for judgement and evaluator is supposed to focus solely on the content that directly addressed the question and determine whether the model's answer aligns with the standard answer.

In our present work, the evaluator judges an answer as correct (i.e. $\mathcal{J}(\mathbf{x}, \mathbf{y}) = 1$ in equation 1) only when model answers the question and the answer matches the standard answer. It is reasonable since model is expected to have been trained on relevant data and therefore should possess the knowledge. Moreover, this knowledge is not long-tailed, but rather high frequency viewed. Additionally, we analyze the responses from different tested models and find that the proportion of evasive answers (e.g. "I don't know") is only approximately 1%. To enhance judgment accuracy, we instruct the evaluator to provide analysis and include several examples in the prompt. GPT4-0125-preview is chosen as the evaluator, achieving a 96% consistency rate with human, thus validating the effectiveness and accuracy of our evaluation process. The prompt used for evaluation is detailed in Appendix B.1.2.

We evaluate 14 popular open and closed source LLMs on FactualBench, including Baichuan series (Yang et al., 2023), Qwen series (Yang et al., 2024; Bai et al., 2023), Llama-3 series (AI@Meta, 2024), Yi (AI et al., 2024), Command-R series (Gomez, 2024a;b), DeepSeek (DeepSeek-AI, 2024), and GPT4 (Achiam et al., 2023). We prioritize the chat/instruct versions of these models. Detailed settings are shown in Appendix C.1 and the results are listed in Table 2. The accuracy of LLMs on our benchmark ranges from 39.11% to 75.62%, indicating that most models still have lacks even in basic factual QA task. Detailed results for each domain and analysis are shown in Appendix D.

### 3.1.3 POTENTIAL IN MODEL

For questions where model response incorrectly, we observe that it can still generate correct answers when allowed greater diversity in its outputs. Taking Baichuan1 as an example, we encourage the model to generate varied answers by increasing the generation temperature. By sampling model's responses 8 times under this condition (which we refer as *high temperature BO8*), as opposed to the standard inference condition (termed *low temperature BO1*). We consider the model to possess relevant information and the ability to utilize it if at least one of the generated answers is correct. As illustrated in Figure 1(a), comparing the accuracy of BO8 and BO1, we find a significant portion of the model's capability have not been fully stimulated, i.e., potential.

### 3.2 SELF-ALIGNMENT WITH MEMORY

Transformer (Vaswani et al., 2017) achitecture LLMs are trained to solve next-word-prediction tasks follows a statistical paradigm (Arora & Goyal, 2023). As probabilistic models, LLMs can generate

Figure 2: We align model with self-generated data on task related to memory utilization.

Table 4: The construction of tuning set.

| Questions | Correctly answered questions | Correct answers |
|---|---|---|
| 24,000 | 15,489 | / |
| 177,714 | **115,798** (SFT1) | 489,357 (SFT2) |

| Questions | Valid questions | DPO pairs |
|---|---|---|
| 24,000 | 12,950 | 96,737 (DPO1) |
| 177,714 | 98,805 | **743,333** (DPO2) |

diverse answers to the same question based on sampling from a distribution, derived from extensive training data. The model's ability to provide correct answer at high temperature indicates a extent of internalization of relevant knowledge. However, an incorrect distribution will result in a high probability of sampling incorrect answers, which can be caused by insufficient or wrong alignment. Since knowledge utilization is a fundamental ability of LLMs (Zhao et al., 2023), LLMs could have a generalized improvement if better alignment is achieved.

To stimulate potential and enhance knowledge utilization ability, we introduce self-alignment with memory. According to Figure 2, we first collect model's high temperature BO8 answers on the training set of FactualBench, a task exclusively evaluate LLM's memory on pre-training knowledge. Then we evaluate answers' correctness, but only a weaker evaluator is needed instead of GPT4. We choose model's self-generated data as tuning labels to prevent the risk of model hallucination exacerbated by fine-tuning on new knowledge (Gekhman et al., 2024) or learning implicit response patterns from external annotations (Kumar et al., 2024). Among BO8 answers, we construct a maximum of 8 pairwise data $(\mathbf{x}, \mathbf{y}_w, \mathbf{y}_l)$ per question, using correct answers as chosen labels and the wrong ones as rejected labels, which can be formulated by the following constraint conditions:

$$\mathbf{x} \sim \mathbb{D}^{\text{train}}; \ \mathbf{y} \sim \pi_{\text{ref}}(\cdot|\mathbf{x}); \ \mathcal{J}(\mathbf{x}, \mathbf{y}_w) = 1; \ \mathcal{J}(\mathbf{x}, \mathbf{y}_l) = 0. \tag{2}$$

In this way, we can quickly generate tuning set $\mathbb{D}^{\text{tuning}}$ containing massive data without human involvement. Then fine-tune the model on this tuning data in preference way, e.g. Direct Preference Optimization (DPO) (Rafailov et al., 2024) for a precise control through bi-directional signals. The DPO loss is defined as follows:

$$\mathcal{L}_{\text{DPO}}(\pi_\theta; \pi_{\text{ref}}) = -\mathbb{E}_{(\mathbf{x}, \mathbf{y}_w, \mathbf{y}_l) \sim \mathbb{D}^{\text{tuning}}} \left[ \log \sigma(\beta \log \frac{\pi_\theta(\mathbf{y}_w|\mathbf{x})}{\pi_{\text{ref}}(\mathbf{y}_w|\mathbf{x})} - \beta \log \frac{\pi_\theta(\mathbf{y}_l|\mathbf{x})}{\pi_{\text{ref}}(\mathbf{y}_l|\mathbf{x})}) \right], \tag{3}$$

where $\pi_\theta$ is the optimal policy, $\pi_{\text{ref}}$ is the policy before optimization. $\sigma$ denotes for sigmoid function, and $\beta$ is a hyperparameter.

The construction of the tuning data is illustrated in Table 4. We randomly select a subset of 24k questions for an early checkpoint and a question is considered to be valid only if it receives both correct and incorrect answers.

## 4 EXPERIMENTS AND ANALYSIS

In this section, we present the training result obtained using our method. Comparing the effectiveness of SFT and DPO, and tuning data from different sources, we observe that: 1) Training on precise factual QA task that related to pre-training knowledge can lead to generalized improvements on other tasks; 2) Training on pairwise data has an advantage over pointwise data in the task; 3) Training effect is enhanced when tuning on self-generated data. For pairwise data, it is crucial that labels from both directions adhere to the same distribution to avoid being hacked easily.

### 4.1 SETUP

We use Baichuan1 as the experimental model. Beside FactualBench, we also choose 6 other benchmarks to evaluate generalized improvements on other factual tasks, helpfulness and general capabilities: TruthfulQA (Lin et al., 2022) for English, HalluQA (Cheng et al., 2023) for Chinese, CMMLU (Li et al., 2023a) for multiple-choice task, HaluEval (Li et al., 2023b) for detection task, AlpacaEval (Li et al., 2023c) for helpfulness, and AlignBench (Liu et al., 2023) for comprehensive abilities. We introduce the details of benchmarking in Appendix C.2.

Traditional SFT (SFT0) and DPO (DPO0) are conducted as baselines, whose labels are directly provided by GPT4 annotations in FactualBench dataset. Besides, we use self-generated data to train

Table 5: Training results after SFT and DPO tuning on Baichuan1. [1]AlpacaEval is calculated on model answers' win rate against the model before training. [2]When calculating Avg $\Delta$, we multiply the score of AlignBench by 10 to align the accuracy of other benchmarks. AlpacaEval shows a relative win rate hence is not concluded in Avg $\Delta$. Red underline indicates an improvement after training; **Bold** font indicates the best performance on the metric; Red font indicates an overall improvement.

| Baichuan1 | chosen/sft label | rejected label | FactualBench(3642)/acc. | TruthfulQA/acc. | HalluQA/acc. | CMMLU/acc. | HaluEval/acc. | AlignBench/score | AlpacaEval/win rate [1] | Avg $\Delta$ [2] |
|---|---|---|---|---|---|---|---|---|---|---|
| Chat | / | / | 48.24 | 30.23 | 32.00 | 48.85 | 50.35 | 5.03 | 50.00 | / |
| SFT0 | dataset | / | 55.86(+7.62) | 21.30(-8.93) | 22.44(-9.56) | 49.58(+0.73) | 12.40(-37.95) | 3.73(-1.30) | 26.65 | -10.18 |
| **SFT1** | BO8 | / | 51.33(+3.09) | 31.46(+1.23) | 30.00(-2.00) | 48.78(-0.07) | **55.73(+5.38)** | 5.04(+0.01) | 37.58 | +1.29 |
| SFT2 | BO8 | / | 52.37(+4.13) | 28.76(-1.47) | 26.44(-5.56) | 50.15(+1.30) | 53.90(+3.55) | 5.03(-0.00) | 31.06 | +0.32 |
| DPO0 | dataset | dataset | 49.08(+0.84) | 28.89(-1.34) | 19.78(-12.22) | 50.70(+1.85) | 54.89(+4.54) | 4.82(-0.21) | 39.07 | -1.40 |
| DPO1 (ours) | BO8 | BO8 | 57.37(+9.13) | 33.78(+3.55) | 38.44(+6.44) | 50.13(+1.28) | 50.63(+0.28) | 5.30(+0.27) | 54.84 | +3.90 |
| **DPO2 (ours)** | BO8 | BO8 | **58.29(+10.05)** | 35.86(+5.63) | **38.89(+6.89)** | **50.92(+2.07)** | 52.05(+1.70) | **5.38(+0.35)** | **63.99** | **+4.97** |
| SFT then DPO | BO8 | BO8 | 54.74(+6.50) | **37.33(+7.10)** | 36.67(+4.67) | 50.72(+1.87) | 54.02(+3.67) | 5.07(+0.04) | 54.53 | +4.03 |
| SFT and DPO | BO8 | BO8 | 57.16(+8.92) | 34.76(+4.53) | 38.22(+6.22) | 50.78(+1.93) | 52.31(+1.96) | 5.13(+0.10) | 63.91 | +4.09 |

| Baichuan1 | Avg. | Professional Knowledge | Mathematics | Fundamental Language Ability | Logical Reasoning | Advanced Chinese Understanding | Writing Ability | Task-oriented Role Play | Open-ended Questions |
|---|---|---|---|---|---|---|---|---|---|
| Chat | 5.03 | 5.34 | 2.71 | 5.57 | 3.20 | 5.86 | 6.32 | 6.33 | 6.63 |
| SFT0 | 3.73 | 4.48(-0.86) | 2.62(-0.09) | 4.79(-0.78) | 2.75(-0.45) | 5.08(-0.78) | 3.24(-3.08) | 3.77(-2.56) | 3.76(-2.87) |
| **SFT1** | 5.04 | 5.78(+0.44) | 2.59(-0.12) | 5.47(-0.10) | 3.30(+0.10) | 5.66(-0.20) | 6.11(-0.21) | 6.25(-0.08) | 6.58(-0.05) |
| SFT2 | 5.03 | 5.46(+0.12) | 2.88(+0.17) | 5.60(+0.03) | 3.25(+0.05) | 5.57(-0.29) | 6.19(-0.13) | 6.17(-0.16) | 6.63(0.00) |
| DPO0 | 4.82 | 4.67(-0.67) | 2.60(-0.11) | 5.53(-0.04) | 3.30(+0.10) | 5.50(-0.36) | 6.40(+0.08) | | 6.00(-0.63) |
| DPO1 (ours) | 5.30 | 5.92(+0.58) | 3.02(+0.31) | 5.66(+0.09) | 3.37(+0.17) | 5.97(+0.11) | **6.53(+0.21)** | **6.55(+0.22)** | 6.79(+0.16) |
| **DPO2 (ours)** | **5.38** | **6.25(+0.91)** | **3.03(+0.32)** | **5.76(+0.19)** | **3.55(+0.35)** | **6.12(+0.26)** | 6.52(+0.20) | 6.36(+0.03) | 6.79(+0.16) |
| SFT then DPO | 5.07 | 5.57(+0.23) | 2.66(-0.05) | 5.53(-0.04) | 3.01(-0.19) | 6.00(+0.14) | 6.33(+0.01) | 6.32(-0.01) | 6.92(+0.29) |
| SFT and DPO | 5.13 | 5.60(+0.26) | 2.79(+0.08) | 5.57(+0.00) | 3.16(-0.04) | 6.05(+0.19) | 6.17(-0.15) | 6.41(+0.08) | **7.16(+0.53)** |

SFT in two settings: one label for each question (SFT1) and all correct answers as labels for each question (SFT2); And we train DPO using our self-alignment with memory in different data sizes (DPO1 and DPO2). Some works claim that fusing DPO loss with SFT loss can help mitigate the overoptimization on rejected labels (He et al., 2024; Liu et al., 2024c), we donate this process as *SFT and DPO*; And additional SFT training before DPO on the preference tuning set can reduce distribution shift issue therefore help training (Xu et al., 2024), which we donate as *SFT then DPO*. More training details are shown in Appendix C.3.

## 4.2 RESULTS

We show the results in Table 5. Our method (DPO2) achieves unanimous improvement on all benchmarks and sub-dimensions of AlignBench, with an average improvement of 4.97% on 6 benchmarks, and a helpful win rate of 63% against Chat model, demonstrating the effectiveness of our method. According to Figure 3, our method stimulates the potential of model, and the improvement is mainly sourced from existing memory rather than new knowledge.

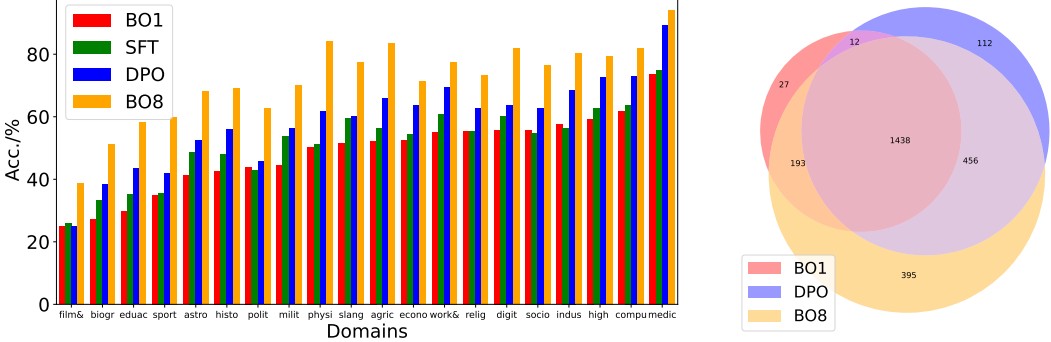

Figure 3: Performance on FactualBench after self-alignment with memory. Left: The accuracy of low temperature BO1, SFT, DPO (ours), and high temperature BO8 on different domains (each domain is represented by its first 5 letters). Right: Venn graph showing the correct answer distribution between low temperature BO1, DPO (ours), and high temperature BO8.

In contrast, traditional training methods that directly use the ground truth annotations exhibit fluctuations in performance on new tasks: hallucinate more on TruthfulQA and HalluQA, and have decline in fundamental abilities and helpfulness. SFT0 even shows a serious decline in instruction

following on HaluEval. Since annotations in dataset are relatively short and concise, they could work as biased signals, which emphasizing the advantage of self-generated data. Comparing SFT and DPO, we find that pairwise data will lead to a greater improvement on training performance (although DPO1 has less tuning data than SFT1), indicating that using unidirectional signal brought by SFT is still insufficient for our task.

We use different sizes of data for DPO training and present the training results in Figure 4. As the size of trainig data increases, the overall improvement, measured by Avg $\Delta$, of the model shows an approximate logarithmic curve, indicating early training on our training set can already effectively improve the model's ability.

In addition, comparing SFT1 and SFT2, we observe that using more correct labels on the same question in SFT doesn't improve the training effect, which limits the effective tuning data size of SFT when the number of training questions is fixed. We also notice that neither fusing SFT loss in DPO loss nor conduct a SFT training before DPO training improves the training results. Since the pairwise data are all sampled from the model itself, there will be little distribution shift and is difficult to overoptimize solely on rejected lables, which also indicates the training robustness of the data constructed by our method.

So far, it still need to be answered why self-alignment with memory effective for mitigating factual hallucination and could lead to generalized improvement.

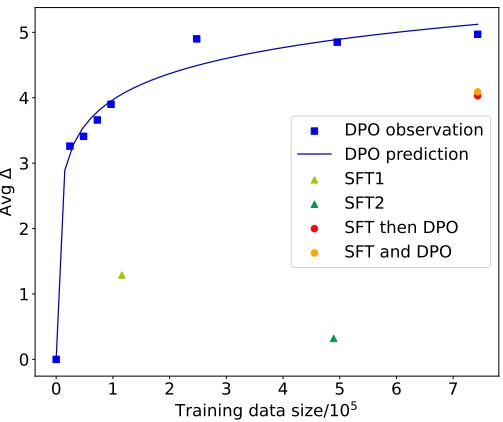

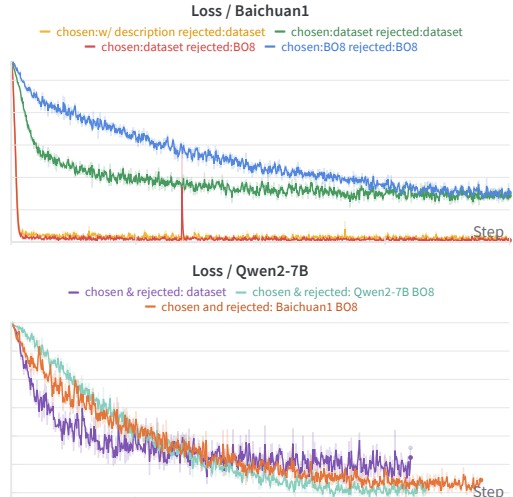

Figure 4: Fitted logarithmic scaling law for training data size of DPO training on Bachuan1. The improvement shows a decreasing marginal benefit trend with the increase of training data size.

Figure 5: Loss comparison between different sources of tuning data.

### 4.3 ANALYSIS

In transformer architecture LLMs, Attention layers and Multilayer Perceptron (MLP) layers extract useful features from input (Jiang et al., 2024), in which MLP layers are regarded to implement a lookup table for factual recall to output multi-token embedding with related information (Nanda et al., 2023) and provide factual knowledge (Dai et al., 2022). Inspired by the skill graph (Arora & Goyal, 2023), we summarize the role of MLP as a memory graph, with edge connecting different input contexts and internalized knowledge. A more precise memory graph leads to better knowledge utilization, and therefore results in accurate answer distribution. The task we choose is highly related to knowledge utilization, providing a direct optimization signal for the ability, a fundamental ability that could affect diverse tasks.

In contrast to the single signal provided in SFT, preference learning like DPO provides a bi-directional signal, which can weaken useless or wrong edge as well as connect or strengthen an needed edge, a more precise control than SFT. Besides, there could be diverse data samples on the same input, making training process more robust. Compared to external label, a self-generate label

has the same answer mode as model itself, avoiding overoptimizing on new answer pattern. And we could also explain why a model fine-tuned on new knowledge it doesn't possess will cause more hallucinations: Since there is no corresponding knowledge in memory graph, model cannot learn how to obtain the correct distribution but an overfitting on the label through building incorrect edges.

A better knowledge utilization leads to better representation. Recent work (Huh et al., 2024) shows that representation alignment increases with LLM's scale and performance. We choose Qwen2-72B-Instruct (Bai et al., 2023) as a good representation function, and the last hidden state value as representation. Our experiments prove that after DPO training, Baichuan1 has further aligned with Qwen2-72B, indicating a better representation ability is achieved. We use mutual nearest-neighbor metrics (Huh et al., 2024) (we explain the calculation process of the metric in the Appendix E) to evaluate alignment between two models, set $k = 10$, and randomly select 200 data points to calculate.

We show the alignment change before and after training based on FactualBench in Figure 6, and alignment changes based on another four benchmarks in Figure 7. Our method achieves higher alignment with Qwen2-72B on four benchmarks and most domains in FactualBench, indicates model has gained stronger representation ability, and result in a better response accuracy. Since different tasks have different difficulties and features, the alignment cannot accurately predict the performance of the model. However, it still works as a signal, reflecting the trend of the model performance changing. Comparing with SFT, DPO has higher alignment degree on average, proving that bi-directional signal of DPO is better for model to achieve generalized improvements.

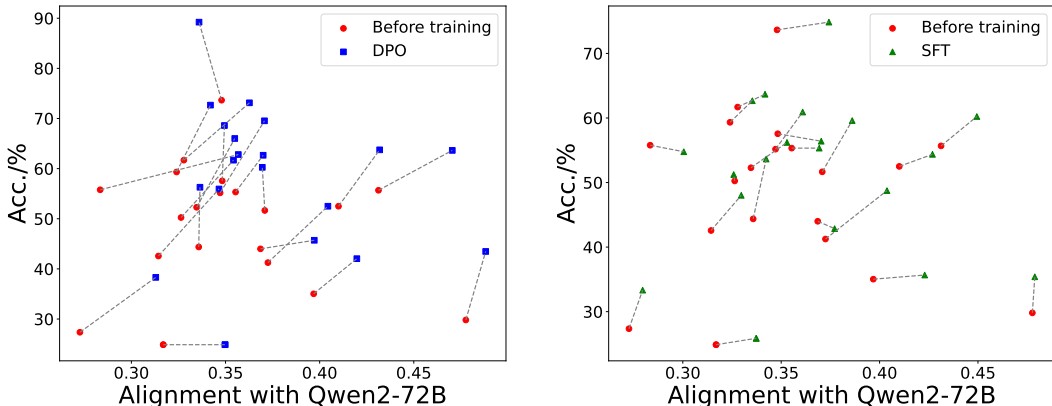

Figure 6: The change of Baichuan1's alignment with Qwen2-72B-Instruct on FactualBench after training. Each point represents a domain. Left: DPO (ours); Right: SFT.

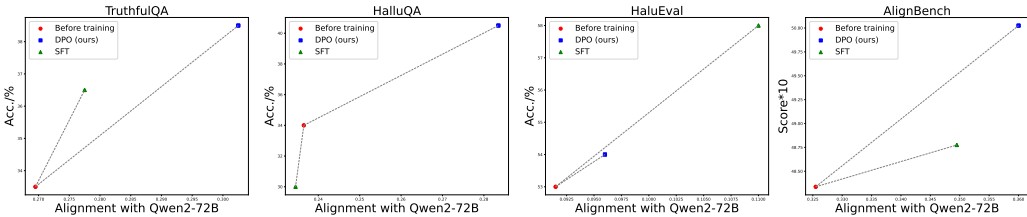

Figure 7: The changes of Baichuan1's alignment with Qwen2-72B-Instruct on four benchmarks after training. From left to right: TruthfulQA, HalluQA, HaluEval, AlignBench. Acc. and score are calculated on selected samples.

## 4.4 ABLATION STUDIES

We conduct several ablation experiments to investigate the impact of different sources of tuning data. Additionally, we experiment with Qwen2-7B-Instruct to validate the effectiveness of our method, but only use the subset of 24,000 questions as early training can already effectively improve model's

Table 6: Training results of ablation studies on Baichuan1 and Qwen2-7B. The default BO8 data are sampling from the model to be trained. BC.BO8 denotes for BO8 sampling from Baichuan1. w/ desc denotes for BO1 with description.

| Baichuan1 | chosen/sft label | rejected | FactualBench/acc. | TruthfulQA/acc. | HalluQA/acc. | CMMLU/acc. | HaluEval/acc. | AlignBench/score | AlpacaEval/win rate | Avg Δ |
|---|---|---|---|---|---|---|---|---|---|---|
| Chat | / | / | 48.24 | 30.23 | 32.00 | 48.85 | 50.35 | 5.03 | 50.00 | / |
| **SFT1** | BO8 | / | 51.33(+3.09) | 31.46(+1.23) | 30.00(-2.00) | 48.78(-0.07) | **55.73(+5.38)** | 5.04(+0.01) | 37.58 | +1.29 |
| SFT3 | w/ desc | / | 55.63(+7.39) | **36.60(+6.37)** | 27.11(-4.89) | **51.39(+2.54)** | 10.40(-39.95) | 4.47(-0.56) | 36.96 | -5.69 |
| SFT0 | dataset | / | 55.86(+7.62) | 21.30(-8.93) | 22.44(-9.56) | 49.58(+0.73) | 12.40(-37.95) | 3.73(-1.30) | 26.65 | -10.18 |
| **DPO2 (ours)** | BO8 | BO8 | **58.29(+10.05)** | 35.86(+5.63) | **38.89(+6.89)** | 50.92(+2.07) | 52.05(+1.70) | **5.38(+0.35)** | 63.99 | **+4.97** |
| DPO3 | w/ desc | BO8 | 18.17(-30.07) | 13.10(-17.13) | 9.33(-22.67) | 48.05(-0.80) | 48.57(-1.78) | 4.07(-0.96) | 32.80 | -13.67 |
| DPO4 | dataset | BO8 | 5.40(-42.84) | 3.92(-26.31) | 1.56(-30.44) | 46.85(-2.00) | 40.10(-10.25) | 3.28(-1.75) | 19.07 | -21.56 |
| DPO0 | dataset | dataset | 49.08(+0.84) | 28.89(-1.34) | 19.78(-12.22) | 50.70(+1.85) | 54.89(+4.54) | 4.82(-0.21) | 39.07 | -1.40 |

| Qwen2-7B | chosen/sft label | rejected | FactualBench/acc. | TruthfulQA/acc. | HalluQA/acc. | CMMLU/acc. | HaluEval/acc. | AlignBench/score | AlpacaEval/win rate | Avg Δ |
|---|---|---|---|---|---|---|---|---|---|---|
| Instruct | / | / | 56.27 | 52.75 | 46.44 | 80.85 | 52.30 | 6.69 | 50.00 | / |
| **SFT4** | BO8 | / | 55.43(-0.84) | 50.31(-2.44) | 45.56(-0.88) | 80.22(-0.63) | 53.70(+1.40) | 6.63(-0.06) | 44.22 | -0.66 |
| SFT5 | BC.BO8 | / | 49.97(-6.30) | 29.87(-22.88) | 24.67(-21.77) | 77.49(-3.36) | 42.05(-10.25) | 4.97(-1.72) | 15.03 | -13.63 |
| SFT6 | dataset | / | 50.38(-5.89) | 19.58(-33.17) | 21.11(-25.33) | 79.85(-1.00) | 9.69(-42.61) | 3.56(-3.13) | 7.20 | -23.22 |
| **DPO5 (ours)** | BO8 | BO8 | **58.81(+2.54)** | **54.47(+1.72)** | **49.78(+3.34)** | **82.15(+1.30)** | 54.00(+1.70) | **6.96(+0.27)** | 58.26 | **+2.22** |
| DPO6 | BC.BO8 | BC.BO8 | 58.17(+1.90) | 53.86(+1.11) | 46.67(+0.23) | 80.14(-0.71) | 52.26(-0.04) | 6.71(+0.02) | 39.19 | +0.45 |
| DPO7 | dataset | dataset | 55.75(-0.52) | 52.14(-0.61) | 46.22(-0.22) | 80.77(-0.08) | 51.70(-0.60) | 6.50(-0.19) | 36.06 | -0.65 |

abilities. In this section, we introduce another data source: *BO1 with description*, which provides reference description to assist in questioning the model under a low temperature configuration. Since the standard answer is contained in description, BO1 with description answers are basically correct, and we use them as chosen/sft label for DPO and SFT training. Despite being generated by the same model, the distribution of BO1 with description answer still differs significantly from the model's own distribution due to the varying input contexts. The settings and the training results are shown in Table 6.

**For both SFT and DPO, self-generated data yield relative better results.** Comparing SFT1, SFT3, and SFT0, although SFT3 and SFT0 exhibit greater improvements on FactualBench, they have significant declines in instruct following (HaluEval) and comprehensive ability (Alignbench). Comparing SFT4, SFT5, and SFT6, the Avg Δ of SFTs all decrease, suggesting that achieving generalized improvements and stimulating a model's potential via SFT is challenging. As for DPO, DPO2 and DPO5 utilizing self-generated data both achieve the best performances and consistent improvements across all benchmarks, underscoring the effectiveness of bi-directional signals.

**For DPO, training data from alternative sources could still yield positive effects. But chosen and rejected labels should be sampled from the same distribution.** Comparing DPO5 and DPO6, which are trained on Qwen2 with data generated by Qwen2 and Baichuan1 both achieve improvements on FactualBench and Avg Δ. However, when chosen and rejected labels are sourced from different distributions (DPO3, DPO4), the training process is quickly hacked, resulting in undesirable parameter changes. According to Figure 5, the loss curve shows a straightly downward trend at the begining, leading to a deterioration of basic conversational abilities, characterized by repetitive and incoherent responses. In cases where chosen and rejected labels are all from dataset, although they are not strictly from the same distribution, it still difficult to be hacked, and therefore there are no significant performance degradation after training.

Detailed training results on FactualBench and AlignBench can be found in Appendix F.

## 5 CONCLUSION

In this article, we aim to mitigate factual hallucination and achieve generalized improvement in model performance. We select factual QA as our training task to enhance model's ability to utilize its memory, i.e. existing knowledge derived from pre-training data. We first extract knowledge from encyclopedia to construct a large-scale, multi-domain Chinese factual QA dataset FactualBench. Based on FactualBench, we observe that model still possesses significant potential for knowledge utilization. Consequently we propose self-alignment with memory: construct self-generated tuning data on FactualBench and train the model using DPO loss. Our method significantly improves model's performance on our benchmark as well as multiple other open-source benchmarks that evaluate factuality, helpfulness, and comprehensive skills. We attribute the effectiveness of our method is originated from better representation ability. Finally, we establish the necessity of bi-directional signals and self-generated data through a series of ablation experiments.

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

APPENDIX

# A    LOW-QUALITY DATA IN PRE-GENERATION

In pre-generation experiment, we randomly sampled 3,000 encyclopedia entries and directly used simple prompts to generate approximately 9,000 questions. After manually evaluating the questions and answers, we found four types of low-quality data: 1) The related knowledge is too long-tailed; 2) The answer to the question is not unique; 3) The answer generated by GPT4 is incorrect; 4) The question is not self-contained.

In this section, we will provide a detailed introduction to these typical low-quality data. In the subsequent generation process, we focused on how to avoid the generation of such low-quality data and formulated a better pipeline. The basic introductions and solutions of four typical types of low-quality data are shown in table 7.

Table 7: Types of low-quality data found in pre-generation and solutions.

| Types of low-quality data | Causes or features | Solutions |
|---|---|---|
| The related knowledge is too long-tailed | The encyclopedia entry is relatively unknown and useless | Filter long-tailed entries based on encyclopedia view counts |
| The correct answer to the question is not unique | The knowledge in encyclopedia entry's description cannot cover all the facts in the world | Use carefully designed prompts for assistance when generating questions, and further filter after generating questions |
| The answer generated by GPT4 is incorrect | The encyclopedia entry's description is too long, which leads to a difficulty in the understanding and instruct following | Filter based on the length of the encyclopedia entry's description |
| | The answer to the question depends on the time limitation (a time-sensitive question) | Use carefully designed prompts for assistance when generating questions, and further filter after generating questions |
| | The encyclopedia entry's description is difficult for model to understand | Low frequency, no additional processing at present |
| The question is not self-contained | The question is not accurate and specific enough | Use carefully designed prompts for assistance when generating questions, and further filter after generating questions |
| | The encyclopedia entry's description itself is not self-contained | Low frequency in entries with high view counts. Filter based on encyclopedia views counts |

**The related knowledge is too long-tailed**    Some entries' subjects are relatively unknown, and the related knowledge appears less on the internet since it is not important for most of people. Questions on those subjects have little value, and are too hard to be answered correctly.

**The correct answer to the question is not unique**    The answers to some questions are not unique and may extend beyond the knowledge provided in encyclopedia entries' descriptions. Additionally, some terms in questions involve subjective judgments hence questions will have more possible and reasonable answers.

**The answer generated by GPT4 is incorrect**    GPT4 is applied to output both question and standard answer based on the description of encyclopedia entry. As the description's length increases, it will challenge model's ability to follow instruction and catch the precise knowledge of the description. Sometimes the description itself is difficult to understand, such as poetry and classical Chinese article, and GPT4 could provide incorrect answers or even semantically incoherent questions facing them. Another situation is that some encyclopedia entries do not emphasize the time limitation of information, and for some rapidly changing data, the knowledge could be outdated.

**The question is not self-contained**    During benchmarking, only the question is provided for the model without encyclopedia content. Therefore, the question should be understandable, complete, and cannot contain vague pronouns or nouns with multiple interpretations such as abbreviations and names without clear context, unless in the vast majority of cases, the word has the same reference or meaning. Sometimes low-quality encyclopedia entries' descriptions themselves are not self-contained.

# B MORE DETAILS ABOUT FACTUALBENCH

## B.1 PROMPTS

In this section, we will provide all prompts related to FactualBench, including generation part and evaluation part.

### B.1.1 GENERATION

**Question generation** (English translation is for reference only):

---

我将提供给你一个对象和相关的参考文档，请针对对象提出最多{提问个数：3}个事实性问题。要求每个问题都具有唯一且准确的答案，避免答案模糊或存在争议，避免涉及主观判断的问题和时效性问题，要求答案可以在参考文档中直接找到。要求提问的问题表达清晰，问题中的名词指代明确，不需要依赖参考文档即可理解问题内容。对每个问题，给出1个标准答案和3个具有干扰性的错误答案。

下面是两个例子：

【对象】：{示例对象1}

【参考文档】：{关于示例对象1的百科内容简介}

【问题1】：{针对示例对象1提出的示例问题1}

【标准答案】：{示例问题1标准答案}

【错误答案1】：{示例问题1错误答案1}

【错误答案2】：{示例问题1错误答案2}

【错误答案3】：{示例问题1错误答案3}

【对象】：{示例对象2}

【参考文档】：{关于示例对象2的百科内容简介}

【问题1】：{针对示例对象2提出的示例问题2}

【标准答案】：{示例问题2标准答案}

【错误答案1】：{示例问题2错误答案1}

【错误答案2】：{示例问题2错误答案2}

【错误答案3】：{示例问题2错误答案3}

对于以下的对象和参考文档，使用同样的格式生成问题、答案。

【对象】：{对象：百科词条对象}

【参考文档】：{文档：百科简介}

---

I will provide you with an object and related reference description. Please generate up to {Question number: 3} factual questions about the object. Each question should have a unique and accurate answer, avoiding vague or contentious answers, subjective judgments, and time-sensitive. The answers should be directly found in the reference description. The questions should be clearly expressed, with unambiguous noun references, and should not rely on the reference description for understanding. For each question, provide one correct answer and three misleading incorrect answers.

Here are two examples:

[Object]: {Example Object 1}

[Reference Description]: {Brief introduction to Example Object 1}

[Question 1]: {Example question 1 related to Example Object 1}

[Correct Answer]: {Standard answer to Example question 1}

[Incorrect Answer 1]: {Incorrect answer 1 to Example question 1}

[Incorrect Answer 2]: {Incorrect answer 2 to Example question 1}

[Incorrect Answer 3]: {Incorrect answer 3 to Example question 1}

[Object]: {Example Object 2}

[Reference Document]: {Brief introduction to Example Object 2}

[Question 1]: {Example question 2 related to Example Object 2}

[Correct Answer]: {Standard answer to Example question 2}

[Incorrect Answer 1]: {Incorrect answer 1 to Example question 2}

[Incorrect Answer 2]: {Incorrect answer 2 to Example question 2}

[Incorrect Answer 3]: {Incorrect answer 3 to Example question 2}

For the following object and reference description, generate questions and answers in the same format.

[Object]: {Object: Encyclopedia Entry Object}

[Reference Description]: {Description: Encyclopedia Description}

---

**Question filtering** (English translation is for reference only):

---

User:你是一个评估专家，下面需要你对一个问题的质量进行判断。

我会给你一个事实性知识问答问题, 你需要从以下几个方面分析这个问题，最终回答问题是【优质】还是【非优质】。

如果这个问题内存在代词指代不清，或无法明确理解问题含义，请回复【非优质】。

如果问题的答案不唯一，请回复【非优质】。

如果问题是时效性问题，且没有给出具体的背景时间点，请回复【非优质】。

如果问题没有以上情况，请回复【优质】。

请一步步思考，并在最后给出你的判断：【优质】或【非优质】。注意将你的最终判断写在中括号【】中！

Assistant:明白了，我会按照你的要求和规则进行判断

User:问题是：

{待评价问题}

请给出你的判断：

---

User: You are an evaluation expert, and you need to assess the quality of a question.

I will provide you with a factual knowledge question, and you need to analyze the question from the following aspects to determine whether the question is of [High Quality] or [Low Quality].

If the question contains unclear pronoun references or cannot be clearly understood, please respond with [Low Quality].

If the answer to the question is not unique, please respond with [Low Quality].

**1026**
**1027** If the question is time-sensitive and does not provide a specific time limitation, please respond with
**1028** [Low Quality].

**1029** If none of the above situations apply, please respond with [High Quality].

**1030**
**1031** Please think through the question step by step and give your final judgment as [High Quality] or
**1032** [Low Quality]. Be sure to put your final judgment in square brackets []!

**1033** Assistant: Understood. I will follow your requirements and rules for evaluation.

**1034** User: The question is:
**1035**
**1036** {Question to be evaluated}

**1037** Please provide your judgment:
**1038**
**1039**
**1040**
**1041**
**1042** B.1.2   EVALUATION
**1043**
**1044** **Answer evaluation** (English translation is for reference only):
**1045**
**1046**
**1047** 给定一个问题以及对应的参考答案，根据参考答案和你掌握的知识，对候选答案是否回答
**1048** 正确进行评价。请注意，问题可能不具有唯一答案，此时只要候选答案言之成理即可。如
**1049** 果候选答案符合参考答案或言之成理，请回答【正确】；如果候选答案与参考答案矛盾或
**1050** 没有回答问题，请回答【错误】，并给出你的分析过程。下面是五个例子：

**1051** 【问题】：百川智能创始人王小川在什么时间与茹立云联合创立了该公司？

**1052** 【参考答案】：百川智能创始人王小川于2023年4月10日与茹立云联合创立了该公司。
**1053**

**1054** 【候选答案】：王小川与茹立云于2023年4月共同创立了百川智能公司。

**1055** 【评价】：根据参考答案，百川智能于2023年4月10日创立，候选答案认为是2023年4月创
**1056** 立，符合参考答案。【正确】

**1057**
**1058** 【问题】：《采桑子·清明后三日作》是哪位诗人创作的？

**1059** 【参考答案】：《采桑子·清明后三日作》是诗人龙榆生创作的。

**1060**
**1061** 【候选答案】：《采桑子·清明后三日作》是清代词人蒋春霖创作的一首词。

**1062** 【评价】：根据参考答案，《采桑子·清明后三日作》是由诗人龙榆生创作，候选答案认为
**1063** 是蒋春霖创作，与参考答案矛盾。【错误】

**1064** 【问题】：李白的代表作有哪些？
**1065**
**1066** 【参考答案】：李白的代表作有《望庐山瀑布》《行路难》《蜀道难》《将进酒》《早发
**1067** 白帝城》《黄鹤楼送孟浩然之广陵》等。

**1068** 【候选答案】：李白的代表作有《将进酒》《静夜思》《庐山谣》《早发白帝城》《赠汪
**1069** 伦》《望庐山瀑布》《行路难》《夜泊牛渚怀古》《登金陵凤凰台》《送友人》等。

**1070** 【评价】：李白有许多代表作，答案不唯一，候选答案中的诗的确均为李白所写，言之成
**1071** 理。【正确】

**1072**
**1073** 【问题】：哈蒂·温斯顿的主要作品有哪些？

**1074** 【参考答案】：哈蒂·温斯顿的主要作品有《灵书妙探第一季》。

**1075** 【候选答案】：哈蒂·温斯顿（Hedy Lamarr）的主要作品有《Ecstasy》（1933年），
**1076** 《Algiers》（1938年），《Samson and Delilah》（1949年）等。
**1077**
**1078** 【评价】：哈蒂·温斯顿有许多作品，答案不唯一，但候选答案中的作品不是哈蒂·温斯顿的
**1079** 作品。【错误】

【问题】：吴之番在哪次战斗中牺牲的？

【参考答案】：吴之番在清顺治二年八月二十六日的战斗中牺牲，这是嘉定三屠的一部分。

【候选答案】：对不起，我找不到关于"吴之番"的相关牺牲信息。这可能是因为您提供的信息有误或者该人物并不存在。

【评价】：根据参考答案，吴之番在顺治二年八月二十六日的战斗中牺牲，候选答案没有回答问题。【错误】

下面是你需要评价的内容，请使用同样的格式给出评价。

【问题】：{问题}

【参考答案】：{参考答案}

【候选答案】：{候选答案}

---

Given a question and its corresponding standard answer, evaluate whether the candidate answer correctly addresses the question based on the standard answer and your knowledge. Please note that the question may not have only one unique answer; in such cases, as long as the candidat answer is reasonable, it is acceptable. If the candidate answer aligns with the reference answer or is reasonable, please respond with [Correct]; if the candidate answer contradicts the reference answer or refuses to answer the question, please respond with [Incorrect] and provide your analysis. Here are five examples:

[Question]: When did Wang Xiaochuan, the founder of Baichuan Inc., co-found the company with Ru Liyun?

[Standard Answer]: Wang Xiaochuan co-founded Baichuan Inc. with Ru Liyun on April 10, 2023.

[Candidate Answer]: Wang Xiaochuan and Ru Liyun co-founded Baichuan Inc. in April 2023.

[Evaluation]: According to the standard answer, Baichuan Inc. was founded on April 10, 2023. The candidate answer states it was founded in April 2023, which aligns with the reference answer. [Correct]

[Question]: Which poet created "Cai Sang Zi · Qing Ming Hou San Ri Zuo"?

[Standard Answer]: "Cai Sang Zi · Qing Ming Hou San Ri Zuo" was created by the poet Long Yusheng.

[Candidate Answer]: "Cai Sang Zi · Qing Ming Hou San Ri Zuo" was created by the Qing Dynasty poet Jiang Chunlin.

[Evaluation]: According to the reference answer, "Cai Sang Zi · Qing Ming Hou San Ri Zuo" was created by Long Yusheng, while the candidate answer claims it was created by Jiang Chunlin, which contradicts the reference answer. [Incorrect]

[Question]: What are the representative works of Li Bai?

[Standard Answer]: Li Bai's representative works include "Wang Lu Shan Pu Bu", "Xing Lu Nan", "Shu Dao Nan", "Qiang Jin Jiu", "Zao Fa Bai Di Cheng", and "Huang He Lou Song Meng Hao Ran Zhi Guang Ling", etc.

[Candidate Answer]: Li Bai's representative works include "Qiang Jin Jiu", "Jing Ye Si", "Lu Shan Yao", "Zao Fa Bai Di Cheng", "Zeng Wang Lun", "Wang Lu Shan Pu Bu", "Xing Lu Nan", "Ye Bo Niu Zhu Huai Gu", "Deng Jin Ling Feng Huang Tai", and "Song You Ren", etc.

[Evaluation]: Li Bai has many representative works, and the answer is not unique. The poems listed in the candidate answer are indeed all written by Li Bai, which is reasonable. [Correct]

[Question]: What are the main works of Hattie Winston?

[Standard Answer]: Hattie Winston's main work is "Castle" (Season one).

[Candidate Answer]: Hedy Lamarr's main works include "Ecstasy" (1933), "Algiers" (1938), and "Samson and Delilah" (1949), etc.

[Evaluation]: Hattie Winston has many works, and the answer is not unique. However, the works listed in the candidate answer are not by Hattie Winston. [Incorrect]

[Question]: In which battle did Wu Zhifan sacrifice?

[Standard Answer]: Wu Zhifan was sacrificed in the battle on August 26, the second year of the Shunzhi reign, which was part of the Jiadin Santu.

[Candidate Answer]: Sorry, I cannot find any information related to Wu Zhifan's sacrifice. This may be due to incorrect information you provided or because this person does not exist.

[Evaluation]: According to the standard answer, Wu Zhifan was sacrificed in the battle on August 26, the second year of the Shunzhi reign, but the candidate answer did not answer the question. [Incorrect]

Here is the content you need to evaluate, and please use the same format to provide your evaluation.

[Question]: {Question}

[Standard Answer]: {Standard Answer}

[Candidate Answer]: {Candidate Answer}

---

### B.2 MORE EXAMPLE

We show one example of each domain from FactualBench in Table 8.

## C DETAILED SETTINGS

In this section, we will introduce the settings of our experiments, including the evlaution and training parts.

### C.1 EVALUATION

We benchmark 14 LLMs on our FactualBench: Baichuan1 (closed), Baichuan2 (open), Qwen1.5-7B-Instruct (open), Qwen2-7B-Instruct (open), Llama-3-8B-Instruct (open), Baichuan3 (closed), Yi-34B-Chat (open), Command-R-35B (open), Llama-3-70B-Instruct (open), Qwen2-72B-Instuct (open), Baichuan4 (closed), Command-R-plus-104B (open), DeepSeek-v2-0628 MoE-236B (open), GPT4-0125-preview (closed), among which DeepSeek and GPT4 are queried from api and others are inferenced locally.

We prioritize the chat/instruct version of the model and use the providing recommended generation config and code on huggingface[2] to generate responses. We set *max_new_tokens* or *max_length* configuration large enough to ensure that models can complete their normal responses.

### C.2 EXPERIMENTS

We choose 6 other open source benchmarks to evaluate model's enhancement comprehensively. We generate answer zero-shot inputing the questions or instructions in default generation config. For model-base evaluation process, we all choose GPT4-0125-preview as evaluator.

**TruthfulQA** Lin et al. (2022) is an English benchmark to measure whether a language model is truthful in generating answers. It contains 817 questions covering 38 domains. The questions are designed to cause imitative falsehoods, a false may due to wrong training objective like fake knowledge in training data. We use the generative part of TruthfulQA and use GPT4 to evaluate the answer (provide reference correct and incorrect answers when judging).

**HalluQA** (Cheng et al., 2023) is a benchmark to measure hallucination phenomenon in Chinese LLM. It contains 450 meticulously designed adversarial questions covering diverse domains to test

---

[2]https://huggingface.co

Table 8: More examples from FactualBench. English translation is for reference only.

| Question | Standard answer | Wrong answer 1 | Wrong answer 2 | Wrong answer 3 | Domain |
|---|---|---|---|---|---|
| 韩国电影《人狼》是由哪位导演执导的? | 电影《人狼》是由金知云执导的。 | 电影《人狼》是由姜栋元执导的。 | 电影《人狼》是由韩孝周执导的。 | 电影《人狼》是由郑雨盛执导的。 | 影视娱乐 |
| 河北师范大学最早起源于哪两所学校? | 河北师范大学最早起源于顺天府学堂和北洋女师范学堂。 | 河北师范大学最早起源于河北师范学院和河北教育学院。 | 河北师范大学最早起源于河北职业技术师范学院和汇华学院。 | 河北师范大学最早起源于北京大学和清华大学。 | 教育培养 |
| 苯丙氨酸的化学式是什么? | 苯丙氨酸的化学式是C9H11NO2。 | 苯丙氨酸的化学式是C8H11NO2。 | 苯丙氨酸的化学式是C9H10NO2。 | 苯丙氨酸的化学式是C9H11NO3。 | 数理化生 |
| 谥号是在什么时期开始的? | 谥号始于西周。 | 谥号始于东周。 | 谥号始于秦朝。 | 谥号始于汉朝。 | 历史国学 |
| 中国电影"第六代导演"之一王小帅的电影女作品是什么? | 王小帅的电影处女作是《冬春的日子》。 | 王小帅的电影处女作是《扁担姑娘》。 | 王小帅的电影处女作是《十七岁的单车》。 | 王小帅的电影处女作是《青红》。 | 人物百科 |
| 法律关系的构成要素有哪些? | 法律关系的构成要素有三项: 法律关系主体、法律关系内容、法律关系客体。 | 法律关系的构成要素有三项: 法律关系主体、法律关系形式、法律关系客体。 | 法律关系的构成要素有三项: 法律关系主体、法律关系内容、法律关系方式。 | 法律关系的构成要素有三项: 法律关系主体、法律关系内容、法律关系目标。 | 政治法律 |
| 国家金融监督管理总局是在哪一年揭牌的? | 国家金融监督管理总局是在2023年揭牌的。 | 国家金融监督管理总局是在2022年揭牌的。 | 国家金融监督管理总局是在2021年揭牌的。 | 国家金融监督管理总局是在2020年揭牌的。 | 经济管理 |
| MemCache是由谁开发的? | MemCache是由LiveJournal的Brad Fitzpatrick开发的。 | MemCache是由Facebook的Mark Zuckerberg开发的。 | MemCache是由Google的Larry Page开发的。 | MemCache是由Microsoft的Bill Gates开发的。 | 计算机科学 |
| 瑞舒伐他汀的主要作用部位是哪里? | 瑞舒伐他汀的主要作用部位是肝。 | 瑞舒伐他汀的主要作用部位是心脏。 | 瑞舒伐他汀的主要作用部位是肾脏。 | 瑞舒伐他汀的主要作用部位是胃。 | 医学 |
| "枫丹白露"这个名字的原义是什么? | "枫丹白露"的法文原义为"美丽的泉水"。 | "枫丹白露"的法文原义为"宏伟的宫殿"。 | "枫丹白露"的法文原义为"狩猎的行宫"。 | "枫丹白露"的法文原义为"古老的城堡"。 | 社会人文 |
| 竹笋原产于哪里? | 竹笋原产于中国。 | 竹笋原产于日本。 | 竹笋原产于印度。 | 竹笋原产于泰国。 | 农林牧渔 |
| 更新世是由哪位地质学家命名的? | 更新世是由英国地质学家莱伊尔命名的 | 更新世是由英国地质学家福布斯命名的 | 更新世是由美国地质学家莱伊尔命名的 | 更新世是由中国地质学家莱伊尔命名的 | 天文地理 |
| 新奥尔良鹈鹕队在哪一年正式宣布球队改名为鹈鹕队? | 新奥尔良鹈鹕队在2013年正式宣布球队改名为鹈鹕队。 | 新奥尔良鹈鹕队在2012年正式宣布球队改名为鹈鹕队。 | 新奥尔良鹈鹕队在2014年正式宣布球队改名为鹈鹕队。 | 新奥尔良鹈鹕队在2015年正式宣布球队改名为鹈鹕队。 | 运动旅游 |
| 宾利汽车公司是在哪一年创办的? | 宾利汽车公司是在1919年创办的。 | 宾利汽车公司是在1920年创办的。 | 宾利汽车公司是在1918年创办的。 | 宾利汽车公司是在1921年创办的。 | 数码汽车 |
| 隔离开关主要用于什么? | 隔离开关主要用于隔离电源、倒闸操作、用以连通和切断小电流电路。 | 隔离开关主要用于调节电压。 | 隔离开关主要用于转换电流。 | 隔离开关主要用于存储电能。 | 工业工程 |
| 鸦片战争是在哪一年开始的? | 鸦片战争是在1840年开始的。 | 鸦片战争是在1842年开始的。 | 鸦片战争是在1839年开始的。 | 鸦片战争是在1841年开始的。 | 军武战争 |
| 买了佛冷这个词是来源于哪首歌曲? | 买了佛冷这个词是来源于歌曲《I Love Poland》。 | 买了佛冷这个词是来源于歌曲《I Love China》。 | 买了佛冷这个词是来源于歌曲《I Love America》。 | 买了佛冷这个词是来源于歌曲《I Love England》。 | 网词网梗 |
| 苏荷酒吧是在哪一年诞生的? | 苏荷酒吧是在2003年诞生的。 | 苏荷酒吧是在2000年诞生的。 | 苏荷酒吧是在2005年诞生的。 | 苏荷酒吧是在2010年诞生的。 | 工作生活 |
| 视觉识别系统VI是什么的缩写? | 视觉识别系统是Visual Identity的缩写。 | 视觉识别系统是Visual Information的缩写。 | 视觉识别系统是Visual Interface的缩写。 | 视觉识别系统是Visual Interaction的缩写。 | 高新科技 |
| 风水业内公认的"龙脉之源"是哪里? | 风水业内公认的"龙脉之源"是昆仑山。 | 风水业内公认的"龙脉之源"是长江。 | 风水业内公认的"龙脉之源"是黄河。 | 风水业内公认的"龙脉之源"是太湖。 | 信仰文化 |
| Who directed the Korean movie 'Inrang'? | The movie 'Inrang' is directed by Kim Jee-woon. | The movie 'Inrang' is directed by Kang Dong Won. | The movie 'Inrang' is directed by Han Hyo-Joo. | The movie 'Inrang' is directed by Jung Woo Sung. | film&entertainment |
| Which two schools did Hebei Normal University first originate from? | Hebei Normal University originated from Shuntianfu Official School and Beiyang Women's Normal School. | Hebei Normal University originated from Hebei Normal Institute and Hebei Institute of Education. | Hebei Normal University originated from Hebei Vocational and Technical Normal College and Huihua College. | Hebei Normal University originated from Peking University and Tsinghua University. | education&training |
| What is the chemical formula for phenylalanine? | The chemical formula for phenylalanine is C9H11NO2. | The chemical formula for phenylalanine is C8H11NO2. | The chemical formula for phenylalanine is C9H10NO2. | The chemical formula for phenylalanine is C9H11NO3. | physics, chemistry, mathematics&biology |
| When did posthumous titles begin? | The posthumous title began in the Western Zhou Dynasty. | The posthumous title began in the Eastern Zhou Dynasty. | The posthumous title began in the Qin Dynasty. | The posthumous title began in the Han Dynasty. | history&traditional culture |
| What is the debut film of Wang Xiaoshuai, one of the "sixth generation directors" of Chinese cinema? | Wang Xiaoshuai's debut film is 'THE DAYS'. | Wang Xiaoshuai's debut film is 'So Close to Paradise'. | Wang Xiaoshuai's debut film is 'Beijing Bicycle'. | Wang Xiaoshuai's debut film is 'Shanghai Dreams'. | biography |
| What are the constituent elements of legal relationships? | There are three elements that make up a legal relationship: the subject of the legal relationship, the content of the legal relationship, and the object of the legal relationship. | There are three elements that make up a legal relationship: the subject of the legal relationship, the form of the legal relationship, and the object of the legal relationship. | There are three elements that make up a legal relationship: the subject of the legal relationship, the content of the legal relationship, and the method of the legal relationship. | There are three elements that make up a legal relationship: the subject of the legal relationship, the content of the legal relationship, and the objective of the legal relationship. | politics&law |
| In which year was the Chinese National Financial Supervisory Administration unveiled? | The Chinese National Financial Supervisory Administration was unveiled in 2023. | The Chinese National Financial Supervisory Administration was unveiled in 2022. | The Chinese National Financial Supervisory Administration was unveiled in 2021. | The Chinese National Financial Supervisory Administration was unveiled in 2020. | economics&management |
| Who developed MemCache? | MemCache was developed by Brad Fitzpatrick from LiveJournal. | MemCache was developed by Mark Zuckerberg from Facebook. | MemCache was developed by Larry Page from Google. | MemCache was developed by Bill Gates from Microsoft. | computer science |
| What is the main site of action of rosuvastatin? | The main site of action of rosuvastatin is the liver. | The main site of action of rosuvastatin is the heart. | The main site of action of rosuvastatin is the kidney. | The main site of action of rosuvastatin is the stomach. | medical |
| What is the original meaning of 'Fontainebleau'? | The original French meaning of "Fontainebleau" is "beautiful spring water". | The original French meaning of "Fontainebleau" is "magnificent palace". | The original French meaning of "Fontainebleau" is "hunting palace". | The original French meaning of "Fontainebleau" is "ancient castle". | sociology&humanity |
| Where do bamboo shoots originate from? | Bamboo shoots originate from China. | Bamboo shoots originate from Japan. | Bamboo shoots originate from India. | Bamboo shoots originate from Thailand. | agriculture, forestry, fisheries&allied industries |
| Which geologist named the Pleistocene epoch? | The Pleistocene was named by British geologist Lyell. | The Pleistocene was named by British geologist Forbes. | The Pleistocene was named by American geologist Lyell. | The Pleistocene was named by Chinese geologist Lyell. | astronomy&geography |
| In which year did the New Orleans Pelicans officially announce their name change to the Pelicans? | The New Orleans Pelicans officially announced their name change to the Pelicans in 2013. | The New Orleans Pelicans officially announced their name change to the Pelicans in 2012. | The New Orleans Pelicans officially announced their name change to the Pelicans in 2014. | The New Orleans Pelicans officially announced their name change to the Pelicans in 2015. | sports&tourism |
| In which year was BentleyMotors Limited founded? | BentleyMotors Limited was founded in 1919. | BentleyMotors Limited was founded in 1920. | BentleyMotors Limited was founded in 1918. | BentleyMotors Limited was founded in 1921. | digital&automotive |
| What is the main use of disconnectors? | DIsconnectors are mainly used for isolating power sources, switching operations, and connecting and disconnecting small current circuits. | Disconnectors are mainly used to regulate voltage. | Disconnectors are mainly used to convert current. | Disconnectors are mainly used for storing electrical energy. | industrial engineering |
| In which year did the Opium War begin? | The Opium War begin in 1840. | The Opium War begin in 1842. | The Opium War begin in 1839. | The Opium War begin in 1841. | military&war |
| What song does the meme 'Mai Le Fo Leng' come from? | The meme 'Mai Le Fo Leng' comes from "I love Poland" | The meme 'Mai Le Fo Leng' comes from "I love China" | The meme 'Mai Le Fo Leng' comes from "I love America" | The meme 'Mai Le Fo Leng' comes from "I love England" | slang&memes |
| In which year was Soho Bar founded? | Soho Bar was founded in 2003. | Soho Bar was founded in 2000. | Soho Bar was founded in 2005. | Soho Bar was founded in 2010. | work&life |
| What word is VI(a Vision System) abbreviation for? | VI abbreviation for Visual Identity. | VI abbreviation for Visual Information. | VI abbreviation for Visual Interface. | VI abbreviation for Visual Interface. | high technology |
| Where is the recognized "source of dragon veins" in chinese feng shui? | The "source of dragon veins" in chinese feng shui is Kunlun Mountain. | The "source of dragon veins" in chinese feng shui is Yangtze River. | The "source of dragon veins" in chinese feng shui is the Yellow River. | The "source of dragon veins" in chinese feng shui is the Taihu Lake. | religion&culture |

models imitative falsehoods and knowledge. Still, we use the generative part and its official prompt to evaluate the answer.

**CMMLU** (Li et al., 2023a) is a Chinese multiple-choice benchmark similar to MMLU Hendrycks et al. (2020), comprising 67 topics and massive questions. We use the official script and code to evaluate model's accuracy by logits output.

**HaluEval** (Li et al., 2023b) is a large collection of generated and human-annotated English hallucinated samples for evaluating the performance of LLMs in recognizing hallucination. It's a discriminative task require test model to judge whether an answer contains hallucination or not. We use the official prompt form to query, and only use 10000 samples from QA part. The evaluation is based on string matching (e.g. "Yes" or "No"), and we have added more matching patterns. If the answer doesn't match any pattern, it will be judge as a wrong answer.

**Alignbench** (Liu et al., 2023) is a Chinese benchmark for evaluating LLMs' alignment. It contains 683 instructions on 8 different fundamental abilities, such as writting, reasoning, role-play, etc. We use its official prompt format to evaluate model's answer in a model-base way.

**AlpacaEval** (Li et al., 2023c) is a benchmark based on the AlpacaFarm (Dubois et al., 2024) evaluation set, which tests model's instrcution following ability. It contains 805 samples on different

instructions, and uses winning rate of the answer against a base model as metric. It has been used to indicate model's helpfulness in previous works (Lin et al., 2024). In our work, the model before training is selected as the based model.

Together with our benchmark FactualBench, we can evaluate model's factuality from generative, multiple-choice, and detective dimensions, as well as Chinese and English language. Besides, we use Alignbench to measure a model's fundamental abilities and AlpacaEval for helpfulness.

### C.3    TRAINING

We conduct training on 32 H800-80G NVIDIA GPU using AdamW optimizer (Loshchilov, 2017). Learning rate is set to be 2e-6 for SFT and 1e-6 for DPO. We use linear scheduler or cosine scheduler with warming up. When fusing DPO with SFT, the weight ratio of DPO loss and SFT loss is 10:1. And we only train one epoch on the tuning set for each method and setting.

## D    DETAILED BENCHMARK RESULTS

We present the performances of 14 LLMs on our FactualBench using a heatmap in Figure 8. The first column represents the overall accuracy of the model and the last line shows the average accuracy of all 14 models. We arrange domains from left to right in descending order of the average accuracy. Each domain is represent by its first 5 letters.

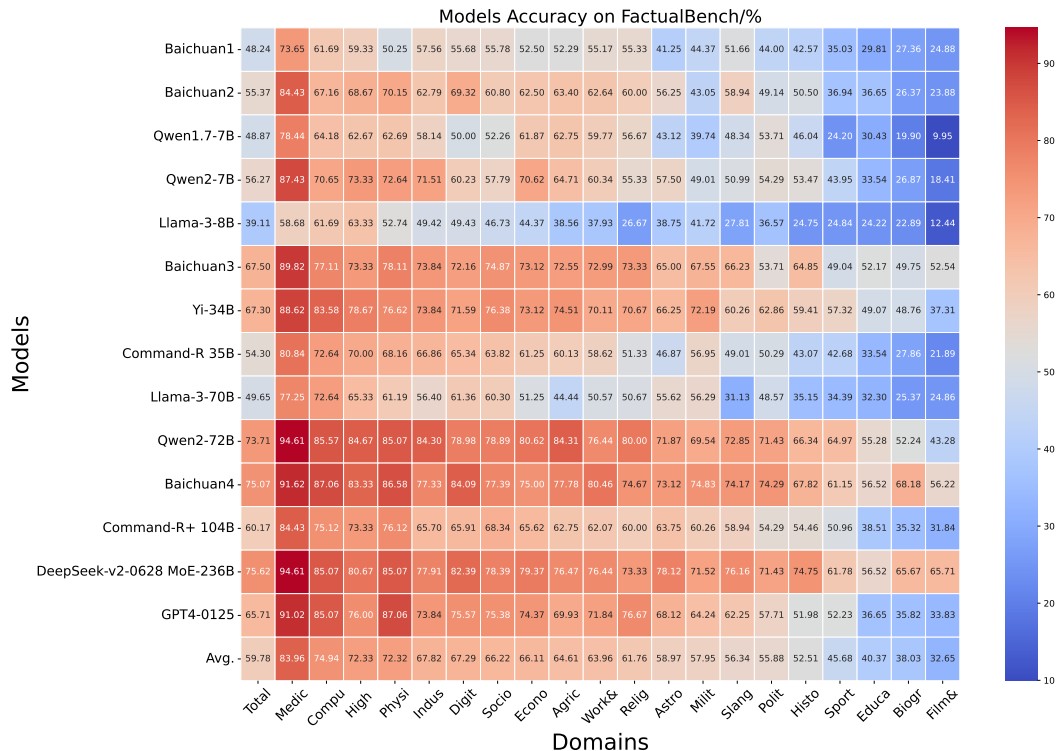

Figure 8: 14 LLMs performances on FactualBench. We prioritize the chat/instruct version.

From Table 9, It can be observed that as models' parameters increase, the accuracy shows an upward trend, while models proficient in Chinese have a significant better performance compared to models proficient in English, which is in line with expectation. Besides, we have two more observations: **1) The same model has significantly different performances on different domains**; **2) Different models share a consistency in relative ability of different domains**, that is, most models' top (bottom) 5 accuracy domains are the same, and no domain exists in both top 5 and bottom 5 domains set at the same time.

Table 9: 14 LLMs performances on FactualBench.

| Model | Proficient lang. | Acc. | Top5 acc. domains(high → low) | Bottom5 acc. domains(low → high) |
|---|---|---|---|---|
| Baichuan1 | CN | 48.24 | medic, compu, high , indus, socio | film&, biogr, educa, sport, astro |
| Baichuan2 | CN | 55.37 | medic, physi, digit, high , compu | film&, biogr, educa, sport, milit |
| Qwen1.5-7B | CN | 48.87 | medic, compu, agric, physi, high | film&, biogr, sport, educa, milit |
| Qwen2-7B | CN | 56.27 | medic, high , physi, indus, compu | film&, biogr, educa, sport, milit |
| Llama-3-8B | EN | 39.11 | high , compu, medic, physi, digit | film&, biogr, educa, histo, sport |
| Baichuan3 | CN | 67.50 | medic, physi, compu, socio, indus | sport, biogr, educa, film&, polit |
| Yi-34B | CN | 67.30 | medic, compu, high , physi, socio | film&, biogr, educa, sport, histo |
| Command-R 35B | EN | 54.30 | medic, compu, high , physi, indus | film&, biogr, educa, sport, histo |
| Llama-3-70B | EN | 49.65 | medic, compu, high , digit, physi | film&, biogr, slang, educa, sport |
| Qwen2-72B | CN | 73.71 | medic, compu, physi, high , agric | film&, biogr, educ, sport, histo |
| Baichuan4 | CN | 75.07 | medic, compu, physi, digit, high | film&, educa, sport, histo, biogr |
| Command-R+ 104B | EN | 60.17 | medic, physi, compu, high , socio | film&, biogr, educa, sport, polic |
| DeepSeek-v2-0628 MoE-236B | CN | 75.62 | medic, physi, physi, digit, high | educa, sport, biogr, film&, polit |
| GPT4-0125-preview | EN | 65.71 | medic, physi, compu, relig, high | film&, biogr, educa, histo, sport |

We demonstrate the phenomenon's occurance is due to two factors. The type of knowledge needed in different domains, and the proportion of data from different domains in the training data, which make domains' task difficulty and LLMs' mastery of domains' knowledge varies. We selected four domains with the poorest performance and four domains with the best performance almost in all 14 models, utilizing all-MiniLM-L6-v2[3] from Sentence Transformer (Reimers, 2019) to extract the features and use t-SNE (Van der Maaten & Hinton, 2008) to reduce the features down to two, visualizing in Figure 9. Different domains questions have significantly different features, comparing the best domains (mainly center at below) and the poorest domains (mainly center at above), indicating the questions distribution varies.

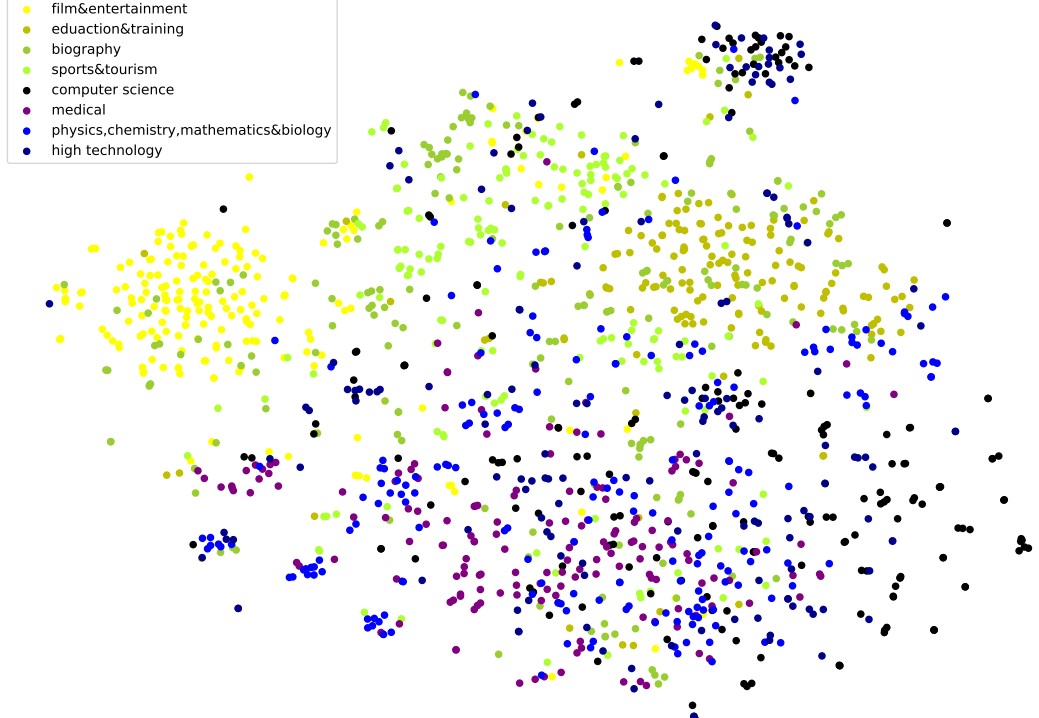

Figure 9: Different domains questions have different features.

[3]https://huggingface.co/sentence-transformers/all-MiniLM-L6-v2

# E   MUTUAL NEAREST-NEIGHBOR METRIC

For two models with representations $f$, $g$, the mutual $k$-nearest neighbor metric measures the average overlap of their respective nearest neighbor sets (Huh et al., 2024). According to the definition (Huh et al., 2024), for each sample $x_i \sim \mathcal{X}, i = 1, 2, ..., b$, two models extract features $\phi_i = f(x_i)$ and $\psi_i = g(x_i)$. The collection of these features are denoted as $\Phi = \{\phi_1, \phi_2, ..., \phi_b\}$ and $\Psi = \{\psi_1, \psi_2, ..., \psi_b\}$. Then we compute the respective nearest neighbor sets $S(\phi_i)$ and $S(\psi_i)$ for each $x_i$ under representations $f$ and $g$.

$$d_{knn}(\phi_i, \Phi\backslash\phi_i) = S(\phi_i) \tag{4}$$

$$d_{knn}(\psi_i, \Psi\backslash\psi_i) = S(\psi_i) \tag{5}$$

where $d_{knn}$ returns the set of indices of its $k$-nearest neighbors. Then we measure its average intersection via

$$m_{\text{NN}}(\phi_i, \psi_i) = \frac{1}{k}|S(\phi_i) \cap S(\psi_i)| \tag{6}$$

where $|\cdot|$ denotes the size of the intersection. We use Euclidean distance to calculate distance betwwen features, and set $b = 200$ (if the size of dataset is less than 200, we sample all data from the dataset), $k = 10$ in our work. The alignment of two representations is measured by $\frac{1}{b}\Sigma_{i=1}^{b}m_{\text{NN}}(\phi_i, \psi_i)$.

# F   DETAILED EXPERIMENT RESULTS

## F.1   ON FACTUALBENCH

Baichuan1 and its related training versions' performances on FactualBench are shown in Figure 10. Qwen2-7B-Instruct and its related training versions' performances on FactualBench are shown in Figure 11.

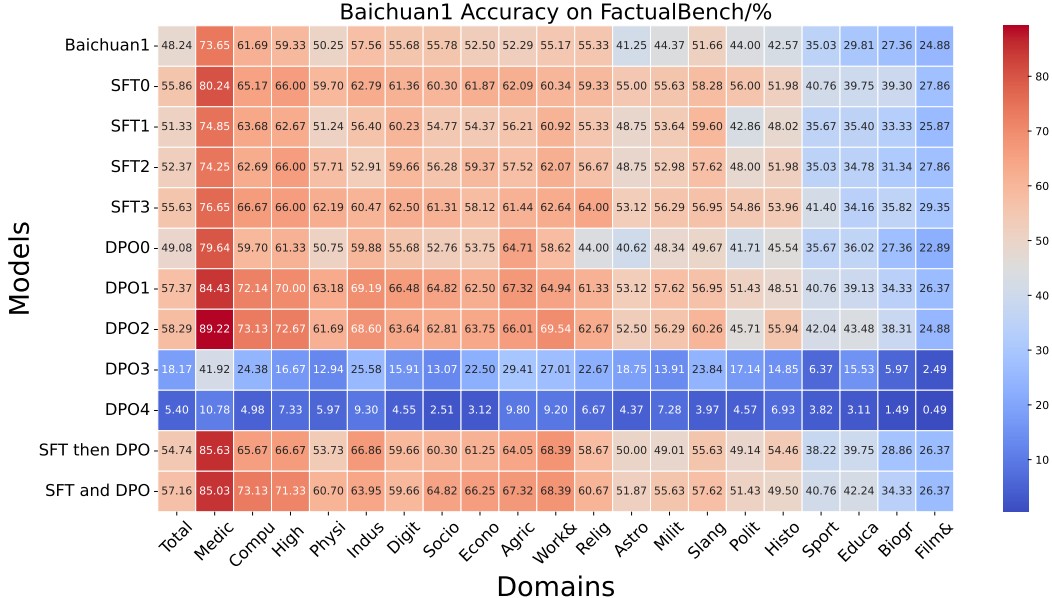

Figure 10: Baichuan1 and its related training versions' performances on FactualBench.

## F.2   ON ALIGNBENCH

Baichuan1 and its related training versions' performances on AlignBench are shown in Table 10. Qwen2-7B-Instruct and its related training versions' performances on AlignBench are shown in Table 11.

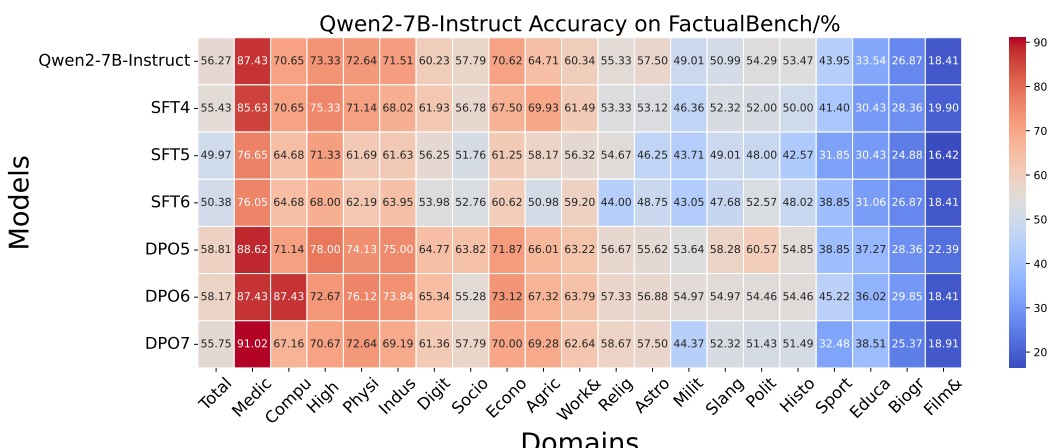

Figure 11: Qwen2-7B-Instruct and its related training versions' performances on FactualBench.

Table 10: Baichuan1 and its related training versions' performances on AlignBench.

| Baichuan1 | Avg. | Professional Knowledge | Mathematics | Fundamental Language Ability | Logical Reasoning | Advanced Chinese Understanding | Writing Ability | Task-oriented Role Play | Open-ended Questions |
|---|---|---|---|---|---|---|---|---|---|
| Chat | 5.03 | 5.34 | 2.71 | 5.57 | 3.20 | 5.86 | 6.32 | 6.33 | 6.63 |
| SFT0 | 3.73 | 4.48(-0.86) | 2.62(-0.09) | 4.79(-0.78) | 2.75(-0.45) | 5.08(-0.78) | 3.24(-3.08) | 3.77(-2.56) | 3.76(-2.87) |
| **SFT1** | 5.04 | 5.78(+0.44) | 2.59(-0.12) | 5.47(-0.10) | 3.30(+0.10) | 5.66(-0.20) | 6.11(-0.21) | 6.25(-0.08) | 6.58(-0.05) |
| SFT2 | 5.03 | 5.46(+0.12) | 2.88(+0.17) | 5.60(+0.03) | 3.25(+0.05) | 5.57(-0.29) | 6.19(-0.13) | 6.17(-0.16) | 6.63(0.00) |
| SFT3 | 4.47 | 5.02(-0.32) | 2.68(-0.03) | 4.96(-0.61) | 2.92(-0.28) | 5.67(-0.19) | 5.32(-1.00) | 5.00(-1.33) | 5.74(-0.89) |
| DPO0 | 4.82 | 4.67(-0.67) | 2.60(-0.11) | 5.53(-0.04) | 3.30(+0.10) | 5.50(-0.36) | 6.40(+0.08) | 6.17(-0.16) | 6.00(-0.63) |
| DPO1 (ours) | 5.30 | 5.92(+0.58) | 3.02(+0.31) | 5.66(+0.09) | 3.37(+0.17) | 5.97(+0.11) | **6.53(+0.21)** | **6.55(+0.22)** | 6.79(+0.16) |
| **DPO2 (ours)** | **5.38** | **6.25(+0.91)** | **3.03(+0.32)** | **5.76(+0.19)** | **3.55(+0.35)** | **6.12(+0.26)** | 6.52(+0.20) | 6.36(+0.03) | 6.79(+0.16) |
| DPO3 | 4.07 | 3.62(-1.72) | 1.93(-0.78) | 4.88(-0.69) | 2.63(-0.57) | 4.47(-1.39) | 5.81(-0.51) | 5.53(-0.80) | 5.34(-1.29) |
| DPO4 | 3.28 | 1.77(-3.57) | 1.95(-0.76) | 4.13(-1.44) | 2.58(-0.62) | 3.71(-2.15) | 5.04(-1.28) | 5.14(-1.19) | 2.55(-4.08) |
| SFT then DPO | 5.07 | 5.57(+0.23) | 2.66(-0.05) | 5.53(-0.04) | 3.01(-0.19) | 6.00(+0.14) | 6.33(+0.01) | 6.32(-0.01) | 6.92(+0.29) |
| SFT and DPO | 5.13 | 5.60(+0.26) | 2.79(+0.08) | 5.57(+0.00) | 3.16(-0.04) | 6.05(+0.19) | 6.17(-0.15) | 6.41(+0.08) | **7.16(+0.53)** |

Table 11: Qwen2-7B-Instruct and its related training versions' performances on AlignBench.

| Qwen2-7B | Avg. | Professional Knowledge | Mathematics | Fundamental Language Ability | Logical Reasoning | Advanced Chinese Understanding | Writing Ability | Task-oriented Role | Open-ended Question |
|---|---|---|---|---|---|---|---|---|---|
| Instruct | 6.69 | 6.62 | 6.65 | 6.51 | 5.06 | 6.76 | 7.15 | 7.59 | 7.46 |
| **SFT4** | 6.63 | **6.74(+0.12)** | 6.40(-0.25) | **7.04(+0.53)** | 4.90(-0.16) | 6.50(-0.26) | 7.09(-0.06) | 7.35(-0.24) | 7.50(+0.04) |
| SFT5 | 4.97 | 5.26(-1.36) | 3.72(-2.93) | 5.88(-0.63) | 3.41(-1.65) | 5.31(-1.45) | 5.60(-1.55) | 5.59(-2.00) | 6.42(-1.04) |
| SFT6 | 3.56 | 4.47(-2.15) | 3.29(-3.36) | 4.50(-2.01) | 3.37(-1.69) | 5.29(-1.47) | 1.92(-5.23) | 2.73(-4.86) | 3.32(-4.14) |
| **DPO5 (ours)** | **6.96** | 6.63(+0.01) | **6.94(+0.29)** | 6.94(+0.43) | **5.56(+0.50)** | 6.93(+0.17) | **7.43(+0.28)** | **7.84(+0.25)** | **7.92(+0.46)** |
| DPO6 | 6.71 | 6.44(-0.18) | 6.37(-0.28) | 6.85(+0.34) | 5.29(+0.22) | **7.26(+0.50)** | 7.21(+0.06) | 7.45(-0.14) | 7.74(+0.28) |
| DPO7 | 6.50 | 6.10(-0.52) | 6.36(-0.29) | 6.76(+0.25) | 4.70(-0.36) | 6.59(-0.17) | 7.23(+0.08) | 7.64(+0.05) | 7.07(-0.39) |

