# OpenReview forum: "Towards Mitigating Factual Hallucination in LLMs through Self-Alignment with Memory"
_ICLR.cc/2025/Conference — ICLR 2025 Conference Withdrawn Submission_

### Official Review · Reviewer_VjHP · 2024-10-26

**Soundness:** 3
**Presentation:** 3
**Contribution:** 2
**Rating:** 3
**Confidence:** 4

**Summary:**

This paper studies the evaluation and mitigation of hallucination in Chinese LLMs. Using QA as the main task, the authors leverage LLMs to generate FactualBench, a large-scale multi-domain benchmark for Chinese question answering. Evaluating a range of Chinese LLMs with the benchmark, the authors argue that the models’ parametric knowledge has the potential to be further unlocked based on the sampling accuracy. A DPO procedure is then introduced to fine-tune Baichuan1, which is shown to be effective based on comprehensive evaluations on seven benchmarks.

**Strengths:**

1. The main contribution of the paper is the synthesized dataset, since collecting the knowledge and creating the questions generally require a large amount of computational cost. With the performance gains from fine-tuning on the proposed dataset, it is promising that the dataset could be used by future studies on LLM factuality in the Chinese LLM space.

2. The empirical evaluation is thorough. A number of LLMs are evaluated on the proposed benchmark. The authors consider seven major benchmarks and different types of training strategies. The empirical results can serve as good reference points for future works on factual alignment.

3. The paper writing is straightforward and the proposed method is clear and intuitive.

**Weaknesses:**

My major concerns about the paper are the data quality and the novelty.

1. **Data Quality**. Based on my understanding, the entire benchmark is generated by different types of LLMs. This limits the paper’s contribution. At least for evaluation, I would argue that high quality human-curated questions are desired.

2. **Misleading Claims**. The claim that only self-generated data is used for alignment is misleading. The main learning signal still comes from the question and the correct answers proposed by GPT-4. On the other hand, the proposed pipeline depends on the correct answers to generate the evaluation signal. If the entire pipeline only involves the LLM being aligned, then it is fair to claim that the data is self-generated.

3. **Novelty**. The proposed approach is essentially an instantiation of [1] in the setting of QA with Chinese LLMs and a self-evaluation method for label generation. Line 124 claims that [1] only studies a single task, but this paper also only focuses on the QA task. In addition, the finding that DPO is more effective with the data directly sampled from the policy is somewhat well-known in the established literature (such as [2]). Therefore, the technical contribution in this paper seems to be below the bar of average ICLR papers.

4. The contribution from this paper is only relevant to Chinese LLMs.

5. There are minor grammatical mistakes throughout the paper (e.g, line 114, 160, 183, 394, 431). I would recommend further proofreading.

[1] Fine-tuning Language Models for Factuality. Tian et al., 2023.

[2] RLHF Workflow: From Reward Modeling to Online RLHF. Dong et al., 2024.

**Questions:**

1. Can you provide details regarding the “encyclopedia” corpus you use? Is it a dump of Chinese websites? Is the dataset published somewhere?

2. Do you have any statistics on the accuracy of the label generation step?

3. What is the size of the training set for DPO1 and DPO2? Also for the other baselines, do they use the same training set size as DPO1, or DPO2?

4. Do you plan to release the data and code?

---

### Official Review · Reviewer_eVmn · 2024-10-28

**Soundness:** 4
**Presentation:** 4
**Contribution:** 4
**Rating:** 8
**Confidence:** 3

**Summary:**

In this article, authors propose a novel benchmark, called FactualBench, for evaluating LLM abilities on the QA taks, generated using articles gathered from Encyclopedia, in Chinese. It contains 181,176 question/answer pairs, on 20 domains (from films to high technology). Authors first evaluate existing LLM on this dataset and other open benchmark, and then propose a novel fine-tuning approach that uses self-generated answers (using high temperature) to improve the LLM capacities on the QA tasks. The proposed approach seems to performs better than strong baselines on all of the datasets.

**Strengths:**

The dataset creation and model’s evaluation are sound and well described

Very interesting experiment of increasing temperature to demonstrate the potential of parametric memory of PLM.
The idea of using self-generated answers as DPO targets is interesting and seems novel.

Comprehensive baseline comparison

Overall, the paper is very clear, provides interesting contributions well motivated and is very well organized/presented.

**Weaknesses:**

The alignment phase might be largely impacted by the judge ability that label answers

Some parts of the paper are not that relevant, such as section 4.3 and beginning of section 3. Some parts of the appendix would be adding more to the body of the article (such as details for baseline implementation)

Minor
Using DPO/SFT-N (1 to 5) makes it hard to read the results tables.

**Questions:**

“We manually rephrase unclear questions to maintain the quality of test set.” Can you provide more details on this manual rephrasing?

Description filtering: it would have been interesting to get the length distribution in appendix. Additionally, this length filtering might affect differently depending on topics and article complexity.

“For questions where model response incorrectly, we observe that it can still generate correct answers when allowed greater diversity in its outputs.”. Could you provide examples? The article could benefit a more robust and formal error analysis.

---

### Official Review · Reviewer_xnpH · 2024-10-29

**Soundness:** 3
**Presentation:** 2
**Contribution:** 2
**Rating:** 5
**Confidence:** 4

**Summary:**

This paper introduces FactualBench, a QA dataset consisting of nearly 200k Chinese QA data spanning 21 domains for both evaluation and training purposes. It also propose self-alignment with memory, fine-tuning the model via preference learning on self-generated pairwise data. The proposed method significantly enhances LLM's performance on FactualBench, and other benchmarks as well.

**Strengths:**

- I love the idea of leveraging existing memory and self-generate labels to mitigate hallucination
- introduce a comprehensive multi-domain Chinese QA dataset
- present an interesting finding that models frequently generate correct answers under high-temperature configurations

**Weaknesses:**

- the figures need some revision, the colors, fonts, captions, and legends are not clear enough eg. instead of SFT1 and SFT2, maybe consider SFT-small and SFT-large
- does not introduce/discuss other related work including [1], which is also about self-alignment on self-generated labels, hallucinations


[1] Self-alignment for factuality: Mitigating hallucinations in llms via self-evaluation. ACL 2024.

**Questions:**

- I'm still somewhat concerned about the assumption that the models have been trained on these encyclopedia data

---

### Official Review · Reviewer_e3mN · 2024-11-06

**Soundness:** 2
**Presentation:** 1
**Contribution:** 2
**Rating:** 1
**Confidence:** 4

**Summary:**

The paper introduces FactualBench, a large-scale, multi-domain Chinese generative QA dataset designed to assess and improve the factuality of large language models (LLMs). The authors propose a self-alignment with memory approach to address LLMs' hallucination problem, which fine-tunes models on self-generated, pairwise data derived from FactualBench. This approach leverages the model's internal knowledge rather than introducing new information. The authors evaluate the tuned LLMs on multiple factuality, helpfulness, and skill-based benchmarks to demonstrate the approach's effectiveness in enhancing factuality and general model performance.

**Strengths:**

1. Curation of FactualBench, a large-scale, multi-domain Chinese generative QA dataset designed to assess the factuality of large language models (LLMs).
2. Comprehensive experiments across datasets to validate their approach
3. 14 language models are benchmarked on FactualBench to show the potential of language models for improved factuality.

**Weaknesses:**

1. Clarity and organization of the paper: The paper’s writing and structure require substantial improvement for clarity and readability. Numerous typos and ambiguous phrases (e.g., "denote" instead of "donate" on page 7) should be revised for accuracy. Additionally, the organization is scattered, with unclear references and sections that make the flow difficult to follow. The main table is cluttered, with small-format numbers that make it challenging to easily follow the trends. Figure 1 does not specify which model is being analyzed. Clear identification of the models used in figures and tables is crucial for interpreting the findings accurately.
2. Questionable interpretation of the Model's internal knowledge: The claim that a model generating a correct answer at high temperatures indicates internalized knowledge is not necessarily valid. High-temperature outputs may include correct answers by chance, as the model could be making educated guesses rather than retrieving stored knowledge. This undermines the assertion that the knowledge exists but is not being utilized effectively.
3. Unclear dataset construction process: The description of the dataset construction process in Section 3.1.1 lacks sufficient detail and clarity. The use of a classifier to categorize data into 20 fixed domains may not adequately capture the full diversity of domains, and alternative methods like dynamic topic modeling might be more effective. Additionally, the criteria for considering generated responses as "low-quality" are vague. A more precise definition of "low-quality" needs to be included in the main text.
4. Inconclusive evidence of the model's knowledge: As mentioned in point 2, generating a correct answer among multiple incorrect ones does not conclusively demonstrate that the model possesses the knowledge. The results in Figure 3 reveal a substantial gap between the performance achieved through DPO training and the model's supposed "true potential," casting doubt on the effectiveness of the proposed method.
5. Limited contributions and lack of novelty: Overall, the paper offers limited contributions. It primarily presents a DPO training pipeline applied to a new dataset without careful consideration and/or explanation of design choices and experimental setup.

**Questions:**

Questions and suggestions are listed below.
1. In Section 3.1.2, it is unclear whether GPT-4 evaluates all data points or only a subsample. If it is the latter, how is the subsample selected?
2. In Section 3.2, why is a weaker evaluator than GPT-4 necessary for assessing answer correctness, and which language model is used as this alternative evaluator?
3. For calculating the average delta, does aggregating scores with different scales make sense? For instance, is multiplying AlignBench scores by 10 and averaging a reasonable aggregation choice?
4. What does "multi-token embedding" refer to in this context: "In transformer architecture LLMs, Attention layers and Multilayer Perceptron (MLP) layers extract useful features from input (Jiang et al., 2024), in which MLP layers are regarded to implement a lookup table for factual recall to output multi-token embedding with related information."
5. Critical experimental and methodological details are missing or moved to the appendix. Please include the necessary information in the main text to facilitate understanding.
6. On the last page, a new data source, "BO1 with description", is introduced. This introduction comes too late for a key component. Please consider restructuring the paper for improved clarity.

---

### Note · Authors · 2024-11-21

I have read and agree with the venue's withdrawal policy on behalf of myself and my co-authors.